# The spatial transcriptomic landscape of the healing mouse intestine following damage

Sara M. Parigi[1,2,7], Ludvig Larsson [3,7], Srustidhar Das[1,2,7], Ricardo O. Ramirez Flores [4], Annika Frede[1,2], Kumar P. Tripathi[1,2], Oscar E. Diaz [1,2], Katja Selin [1,2], Rodrigo A. Morales [1,2], Xinxin Luo[1,2], Gustavo Monasterio[1,2], Camilla Engblom [5], Nicola Gagliani [1,2,6], Julio Saez-Rodriguez [4], Joakim Lundeberg [3] & Eduardo J. Villablanca [1,2✉]

The intestinal barrier is composed of a complex cell network defining highly compartmentalized and specialized structures. Here, we use spatial transcriptomics to define how the transcriptomic landscape is spatially organized in the steady state and healing murine colon. At steady state conditions, we demonstrate a previously unappreciated molecular regionalization of the colon, which dramatically changes during mucosal healing. Here, we identified spatially-organized transcriptional programs defining compartmentalized mucosal healing, and regions with dominant wired pathways. Furthermore, we showed that decreased p53 activation defined areas with increased presence of proliferating epithelial stem cells. Finally, we mapped transcriptomics modules associated with human diseases demonstrating the translational potential of our dataset. Overall, we provide a publicly available resource defining principles of transcriptomic regionalization of the colon during mucosal healing and a framework to develop and progress further hypotheses.

[1] Division of Immunology and Allergy, Department of Medicine Solna, Karolinska Institute and University Hospital, Stockholm, Sweden. [2] Center of Molecular Medicine, Stockholm, Sweden. [3] Science for Life Laboratory, Department of Gene Technology, KTH Royal Institute of Technology, Stockholm, Sweden. [4] Heidelberg University, Faculty of Medicine, and Heidelberg University Hospital, Institute for Computational Biomedicine, Bioquant, Heidelberg, Germany. [5] Department of Cell and Molecular Biology, Karolinska Institute, Stockholm, Sweden. [6] I. Department of Medicine and Department of General, Visceral and Thoracic Surgery, University Medical Center Hamburg-Eppendorf, Hamburg, Germany. [7] These authors contributed equally: Sara M. Parigi, Ludvig Larsson, Srustidhar Das. ✉email: eduardo.villablanca@ki.se

The intestine is divided into the small and large bowels that together host the highest density of commensal microbiota, which in turn is spatially heterogeneous across the proximal-distal axis[1]. The geographically heterogeneous microbial exposure has contributed to the establishment of a highly compartmentalized organ that has distinct functions depending on the proximal-distal location[2,3]. For example, vitamin A-metabolizing enzymes and consequently retinoic acid production and function are higher in the proximal compared to distal small intestine[4], generating a proximal-to-distal gradient. Although it is broadly accepted that the small intestine is highly compartmentalized, whether a clear molecular regionalization exists in the colon is yet to be determined.

The intestine relies on the constant regeneration of the intestinal epithelium to maintain homeostasis. Breakdown in regenerative pathways may lead to pathogen translocation and the development of chronic intestinal pathologies, such as inflammatory bowel disease (IBD)[5]. Therefore, the intestinal barrier must quickly adapt to promote tissue regeneration and healing following injury. However, the cellular and molecular circuitry at steady state conditions and how it adapts upon challenge is yet to be fully characterized.

The intestine offers a unique opportunity to investigate common principles of tissue repair at the barrier because of its spatial organization, which is fundamental to its function. When intestinal barrier injury occurs, damaged epithelial cells are shed and replaced by mobilizing intestinal stem cell (ISC)-derived cells[6], a phenomenon highly dependent on signals coming from the neighboring microenvironment (niche)[7]. Similarly, immune cells are recruited or expanded in situ to protect the host from invading pathogens and to orchestrate the healing process by providing resolving signals[8]. Although initially considered as a mere structural support, stromal cells, which includes fibroblasts, endothelial/lymphatic cells, pericytes and glial cells, are also actively involved in barrier healing through tissue remodeling, matrix deposition, neoangiogenesis, muscle contraction, and production of pro-regenerative signals[9]. Therefore, immune, epithelial and stromal cells must quickly adapt within a defined microenvironment and establish a molecular network to promote tissue repair. However, whether different segments of the intestine and their microenvironments possess distinct types of tissue repair mechanisms is currently unknown.

Our previous study unveiled the temporal transcriptomic dynamic of the colonic tissue over the course of dextran sodium sulfate (DSS) colitis, identifying genes and pathways differentially modulated during acute injury or regeneration[10]. Although bulk or single cell RNA sequencing studies provide unbiased transcriptome analysis, the spatial context within the tissue is typically lost. In contrast, targeted technologies for spatial gene expression analysis (e.g. in situ RNA-sequencing, fluorescence in situ hybridization [FISH], RNA-scope) require knowledge of specific candidate genes to interrogate and thus do not allow an unsupervised investigation of pathways enriched in healing areas.

In this work, we overcome these limitations and we exploit spatial transcriptomics (ST), an unbiased technology allowing sequencing of polyadenylated transcripts from a tissue section, which can be spatially mapped onto the histological brightfield image[11]. ST allowed us to uncover an unprecedented view of the molecular regionalization of the murine colon, which is further validated on human intestinal specimens. By comparing ST of colonic tissue under steady state and upon mucosal healing (i.e. from DSS-treated mice), we identify and spatially map transcriptional signatures of tissue repair processes, immune cell activation/recruitment, pro-regenerative pathways, and tissue remodeling. The spatial landscape of pathway activity during mucosal healing also unveils a negative correlation between p53

activity and proliferating epithelial stem cells. Moreover, targeted mapping of genes associated with disease outcome in human IBD patients and IBD risk variants identified from genome-wide association studies (GWAS) allowed us to infer their involvement in specific pathological processes based on their localization in areas with distinct histological properties.

## Results

**Spatial transcriptomics revealed distinct molecular regionalization of murine colonic epithelium at steady state condition.** To characterize the transcriptomic landscape of the colon tissue at steady state condition, we processed frozen colons for ST using the Visium (10X Genomics) platform (Fig. 1a). The pre-filtered dataset corresponded mostly to protein coding genes (Supplementary Fig. 1a). Upon filtering out non-coding RNAs (ncRNAs) and mitochondrial protein coding genes, the resulting dataset consists of 2604 individual spots, with an average of ~4125 genes and ~11801 unique transcripts per spot (Supplementary Fig. 1b). First, we deconvolved the spatial transcriptomic dataset using non-negative matrix factorization (NNMF) to infer activity maps[12], and we restricted the analysis to only 3 factors that capture the most basic structure of the colon at steady state conditions (d0) (Fig. 1b). We identified 3 basic structural transcriptomic landscapes that were histologically discernible as intestinal epithelial cells (IEC) (NNMF_3), a mixture between lamina propria (LP) and IEC (NNMF_2), and muscle (NNMF_1), which were indistinguishable from the IEC towards the most distal colon (Fig. 1b). Analysis of the top contributing genes for each factor confirmed the identity of the muscle and IEC, and the mixed signature between IEC, muscle, and LP (Fig. 1c). Using immunohistochemistry (IHC) data from the human protein atlas[13], we validated the specific expression of CDH17 (also known as liver-intestine cadherin or LI cadherin)[14] and TAGLN (transgelin, smooth muscle marker) in the IEC and muscularis layer, respectively (Fig. 1d). By contrast, ADH1 (alcohol dehydrogenase 1) showed a mixed expression between the LP and IEC compartments (Fig. 1d). To better visualize the molecular regionalization both across the proximal-distal and serosa-luminal axis, we digitally unrolled the colon (Supplementary Fig. 2a and Fig. 1e) as described in methods. In line with the ST expression in Fig. 1c, the muscle, LP/IEC and proximal IEC factors were enriched in the corresponding regions of the distal-proximal and serosa-luminal axis of the digitally unrolled colon (dark dots in Fig. 1e). Among the genes driving factor 3 (NNMF_3) we found *Car1, Mettl7b, Emp1, Fabp2,* and *Hmgcs2* that were highly expressed in the proximal colon (Fig. 1f and Supplementary Fig. 2b). Our data aligns with reports showing *Car1* promoter-driven expression in the proximal but not distal colon[15]. In contrast, *Retnlb, Sprr2a2* and *Ang4* were enriched in the mid colon, and *Prdx6, Tgm3, Ly6g, Eno3, and B4galt1* were enriched in the distal colon (Fig. 1f and Supplementary Fig. 2b). Because they have not been previously described as markers for the distinct colonic compartments, we used qPCR to validate the region-specific mRNA expression of genes coding for the ketogenic rate-limiting enzyme, mitochondrial 3-hydroxy-3-methyl-glutaryl-CoAsynthase 2 (HMGCS2), the antimicrobial peptide angiogenin 4 (ANG4), and the Beta-1,4-galactosyltransferase 1 (B4GALT1)(Fig. 1g). Therefore, ST analysis distinguished a stratification between the LP, IEC, and muscle compartment which is clearly evident at the proximal but not distal colon.

Some studies have shown structural and functional differences between proximal and distal colon, specifically with respect to the epithelium;[15–20] however, systematic and unsupervised molecular regionalization of the colon is lacking. To objectively identify genes differentially expressed in specific compartments of the colon, we

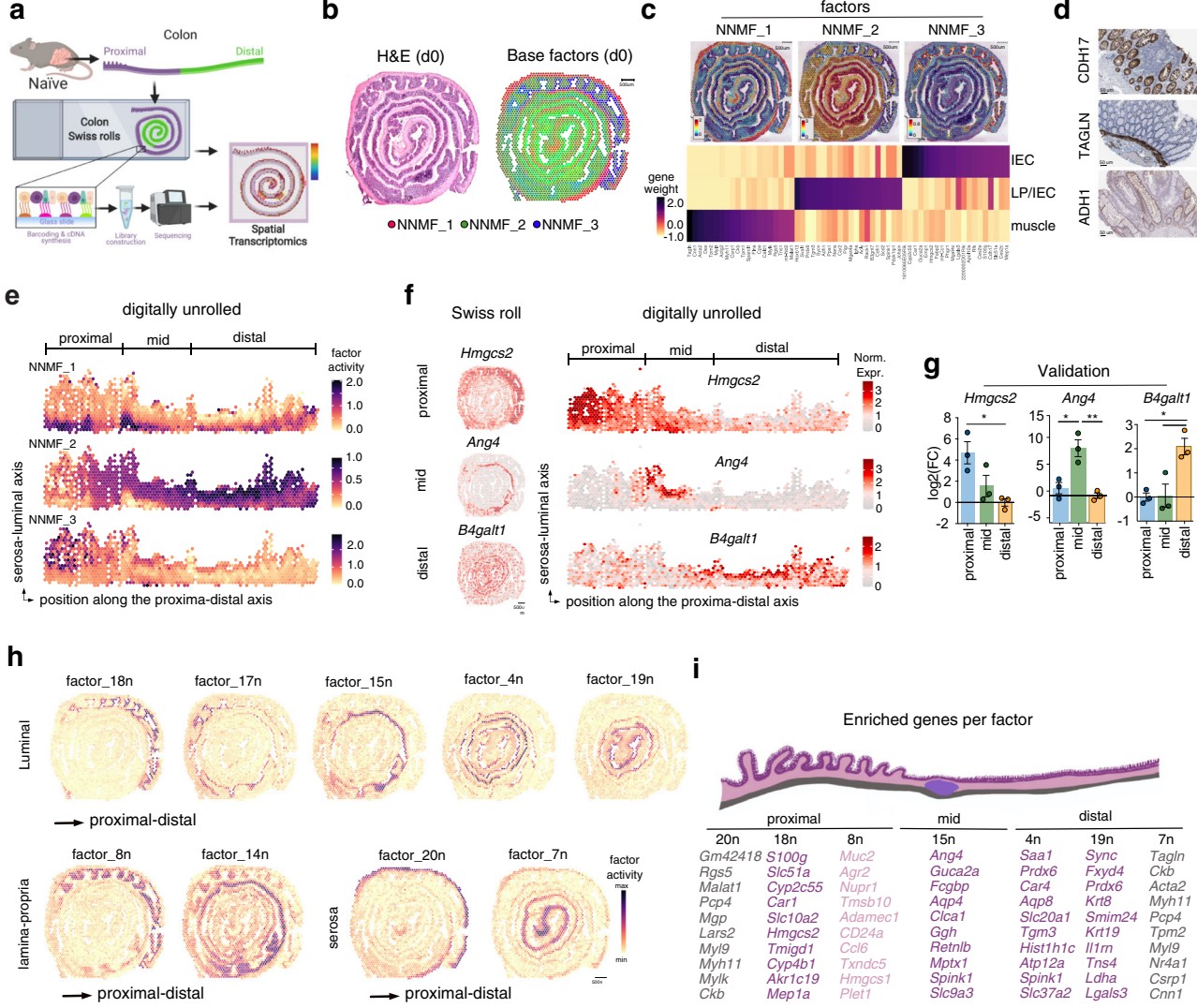

**Fig. 1 Spatial transcriptomics reveals molecular regionalization of the murine colonic tissue in steady state. a** Schematics of the experiment: colonic tissue from a naive WT mouse (d0) was processed as a Swiss roll for spatial transcriptomic (ST) with Visium 10X technology (n = 1 Swiss roll). **b** Colon Swiss rolls shown in hematoxylin and eosin (H&E) staining (left) and with each ST spot color coded based on non-negative matrix factorization (NNMF) (right). ST spots belonging uniquely to one factor are colored in red, blue and green for NNMF1, 2 and 3 respectively. ST spots shared between different factors are colored with respective intermediate gradation of these 3 colors. **c** Top: spatial distribution of the 3 factors distinguishing muscle, lamina propria (LP) and intestinal epithelial cells (IEC). Bottom: heatmap showing the top 20 genes defining each factor. **d** Immunohistochemical staining of CDH17, TAGLN, ADH1 in healthy human colonic tissue (from Human Protein Atlas). **e** Digitally unrolled colonic tissue, showing the distribution of the 3 factors from Fig. 1c along the serosa-luminal and proximal–distal axis. **f** Proximal to distal distribution of *Hmgcs2, Ang4* and *B4galt1* expression in colonic swiss rolls (left) and digitally unrolled colon (right). **g** qPCR validation of regional expression of *Hmgcs2, Ang4* and *B4galt1* in proximal, mid and distal colonic biopsies from WT mice (n = 3, each dot represents one mouse). Data are presented as mean values ± SEM. Significance was assessed by one-way ANOVA with Bonferroni post-test. *$p < 0.05$; **$p < 0.01$. **h** Spatial distribution of 9 out of 20 factors in the naive colon displaying transcriptional regionalization along the serosa-luminal and proximal-distal axis. Each ST spot is assigned a color-coded score based on the expression of the genes defining each factor. **i** Schematic representation of the colon (top) and top genes annotated in Factors. Factors are grouped based on their proximal-distal distribution and color-coded (i.e. gray-pink-purple) based on their serosa-luminal distribution.

used the NNMF method to distinguish relevant sources of variability of the data[12]. Detailed factor analyses of the naive colon (denoted with "n") resulted in a more pronounced/apparent colonic compartmentalization, in which the top genes defining a factor were sufficient to specifically demarcate the regionalization in the colon (Supplementary Fig. 3a, b). In particular, these factors defined a proximal-distal and a serosa-luminal axis (Fig. 1h). Functional enrichment analysis using the top contributing genes of factors defining the proximal and distal colonic IEC suggested that the murine proximal colon is specialized in water absorption, while the distal colon is specialized in solute transport (Supplementary Fig. 3c),

indicating functional differences between the proximal and distal IEC compartments. Altogether, ST permitted the identification of a previously unappreciated level of colonic molecular compartmentalization in the steady state colon, as summarized in Fig. 1i.

**Visualization of lymphoid structures by factors enriched with B-cell associated genes.** Next, we examined the capacity of our dataset to resolve macroscopic structures, such as lymphoid clusters, within the tissue. We observed an enrichment of B cell-associated genes in factors 1n, 3n and 9n (Fig. 2a). Upon mapping these factors onto the colonic tissue, we observed that factors 1n

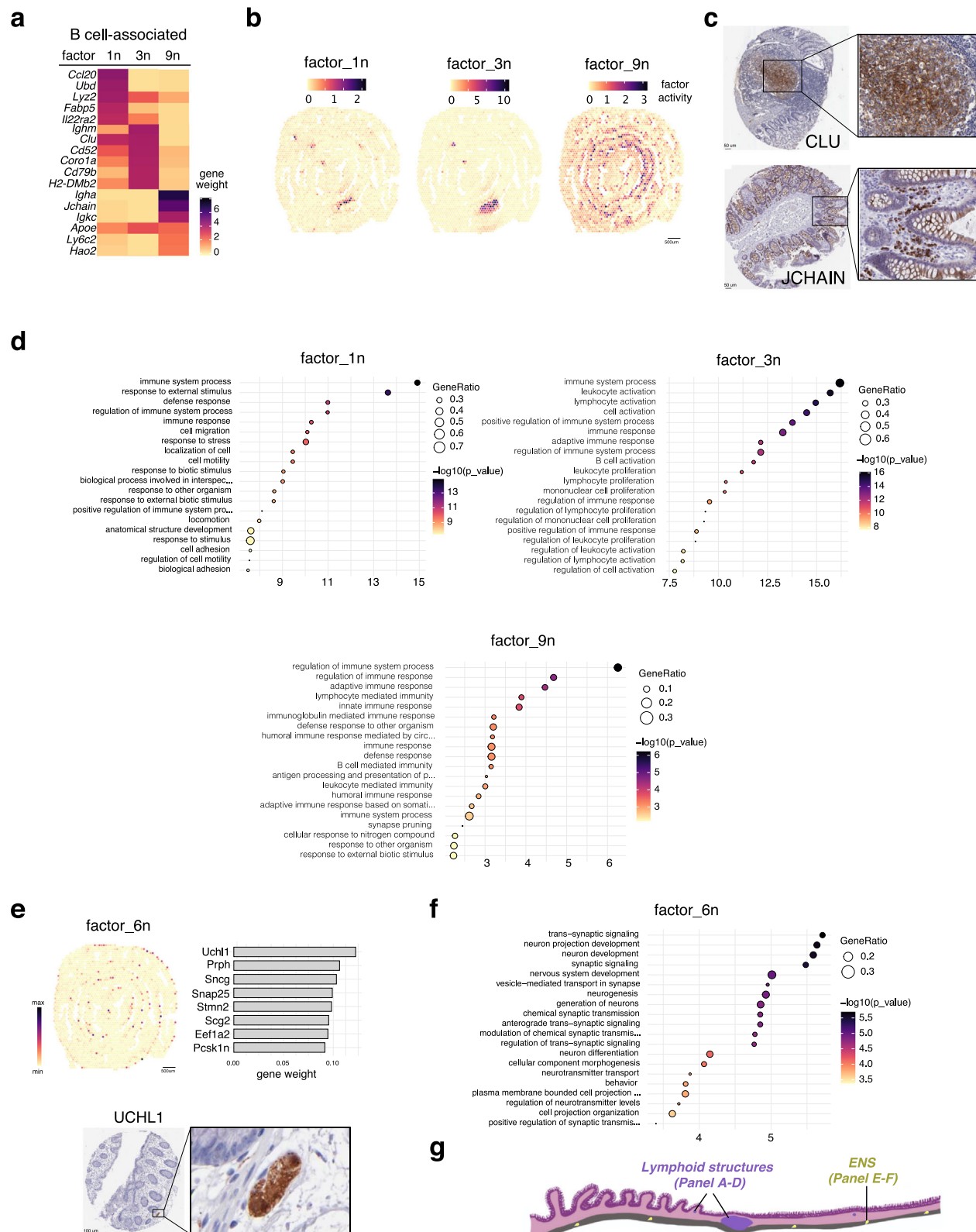

**Fig. 2 Identification and regional distribution of lymphoid follicle, B cell-associated, and enteric nervous system signatures in the naive murine colon.**
**a** Heatmap of the top genes defining factors 1n, 3n and 9n enriched in B cell signature. **b** Spatial distribution of B cell-associated factors in the naive colon.
**c** Immunohistochemical staining of CLU (enriched in lymphoid follicles) and JCHAIN (localized in the lamina propria) in healthy human colonic tissue (from
Human Protein Atlas). **d** Functional enrichment analysis (Gene Ontology, GO) of factors 1n, 3n and 9n. **e** Top: spatial distribution (left) and top genes
(right) defining factor 6n (enteric nervous system, ENS). Bottom: immunohistochemical staining of UCHL1 (neuronal marker) in the colonic submucosa of
healthy human colonic tissue (from Human Protein Atlas). **f** Pathway analysis (GO) of factor 6n. **g** Schematic representation of spatial distribution of B cell
factors (i.e. 1n, 3n and 9n from Panels A-D) and ENS-factor 6n (from Panel E-F).

and 3n defined structures that resembled lymphoid aggregates, known as isolated lymphoid follicles (ILF) and/or cryptopatches (CP) (Fig. 2b). In contrast, factor 9n defined the colonic LP and was characterized by high expression of genes such as *Igha*, *Jchain*, *Igkc* characteristic of plasma cells (Fig. 2b). Among top-listed genes found in factor_3n, we validated *Clu* protein expression in lymphoid follicles[21] (Fig. 2c). Similarly, expression of JCHAIN, a small 15 kDa glycoprotein produced by plasma cells that regulates multimerization of secretory IgA and IgM and facilitates their transport across the mucosal epithelium[22], was validated by IHC in the human colonic LP (Fig. 2c). Pathways analysis confirmed that these factors were associated with immune responses (Fig. 2d). Whereas pathways associated with factor 3n suggested sites of lymphocyte priming (defined by processes associated with lymphocyte activation), pathways associated with factor 9n suggested sites of effector immune responses (defined by processes associated with adaptive immunity) (Fig. 2d). Interestingly, factor_1n, which defined the cells/region overlying the ILF, was characterized by genes such as *Ccl20* known to recruit CCR6$^+$ B cells[23,24] and *Il22ra2/Il22bp*, a soluble receptor that neutralizes the effects of IL-22, a pleiotropic cytokine primarily expressed by lymphoid tissue inducer cells (LTis)[25]. In addition, factor_1n was associated with pathways involved in cell mobilization and response to external stimulus (Fig. 2d). Because of the observation that factor 1n defines ILFs, it is tempting to propose that factor 1n-defined structures may serve as an anlagen for further maturation into ILF (factor_3n).

**Factor analysis identified molecular signatures that define areas associated with the enteric nervous system**. Factor 6n was characterized by an enrichment of enteric nervous system (ENS)-associated genes which were located in the muscle area (Fig. 2e). Among the top-listed genes, we validated ubiquitin C-terminal hydrolase L1 (UCHL1) (Fig. 2e), which is specifically expressed in neurons[26]. Functional enrichment analysis confirmed that factor_6n defined a transcriptomic profile associated with the ENS (Fig. 2f). In summary, the resolution of our Visium dataset permitted the identification of known structures (e.g. ENS and ILFs; Fig. 2g), thereby providing a platform to further investigate specific molecular circuitry within such regions.

**Molecular landscape of intestinal mucosal healing**. Next, we sought to spatially resolve the colonic transcriptomic landscape during mucosal healing. We took advantage of our recent work showing that by day (d)14, the intestinal barrier integrity is restored following damage induced by dextran sodium sulfate (DSS)[10]. Therefore, we treated wild type (WT) mice with DSS in drinking water for 7 days followed by 7 days of recovery and d14 colonic tissue was taken to generate frozen Swiss-rolls to be processed for ST (Fig. 3a). Despite the recovery at a physiological level (i.e. body weight gain) (Supplementary Fig. 4a), the colonic tissue after DSS treatment did not fully return to homeostasis, as demonstrated by its reduced length (a sign of inflammation) (Supplementary Fig. 4b).

At the histological level, large lymphoid patches, as well as the muscle and mucosal layer across the intestine, were easily identified (Supplementary Fig. 4c). Hematoxylin and eosin (H&E) sections annotated by a blinded pathologist revealed the heterogeneity of the tissue, including the presence of isolated lymphoid follicles (ILFs), as well as areas with edema, hyperplasia, crypt duplication, and normal tissue (Supplementary Fig. 4d). Of note, the distal colon (center of the Swiss roll) showed marked alterations, whereas the proximal colon (outer Swiss roll) seemed non-affected (Supplementary Fig. 4d).

The d14 ST dataset consisted of 3630 individual spots with a number of unique genes per spot (nFeature_RNA) that was comparable to the d0 tissue section (Supplementary Fig. 1a). To first appreciate how the process of mucosal healing spatially altered the colonic transcriptome, the ST data from d0 and d14 were embedded in 3 dimensions using Uniform Manifold Approximation and Projection (UMAP). The values of these 3 dimensions were then re-scaled into a unit cube (with a range of 0 to 1) and used as channels in CMYK color space to generate a specific color for each ST spot (Fig. 3b, bottom part). Interestingly, lymphoid follicles (identified by H&E staining) and areas in the proximal colon showed high similarity (i.e. same color) between d0 and d14 samples (Fig. 3c), suggesting that these structures are transcriptionally less affected during the process of mucosal healing following intestinal injury. Vice versa, in line with the histo-pathological scoring, the distal portion of the d14 colon was the most dramatically affected region.

To visualize how the colonic tissue is transcriptionally organized in different areas, we integrated the data from d0 and d14 using harmony[27] and performed cluster analysis. We annotated 17 distinct clusters, which were visualized by embedding the data in 2 dimensions with UMAP (Fig. 3c). Differentially up-regulated genes per cluster are summarized in a heatmap showing the top conserved genes in each cluster (Fig. 3d). Each cluster defined a distinct geographic area of the tissue (Supplementary Fig. 5). For instance, cluster 12 designated the ENS, with scattered expression in the submucosal layer, whereas cluster 0 mapped spatially to the proximal colon (Fig. 3e). Interestingly, genes defining cluster 0 expanded towards the mid colon during mucosal healing (d14) (Fig. 3e). Among these genes, *Muc2* and *Reg3b* showing expanded expression towards the mid colon (Fig. 3f and supplementary Data File 1) play a key role in establishing the barrier integrity. Using qPCR, we validated the expanded expression of *Reg3b* during mucosal healing (Fig. 3g). Overall, cluster analysis revealed that despite the existence of a conserved transcriptional colonic regionalization, the process of tissue healing underlies the emergence of distinct molecular signatures and alters the distribution of specific gene expression.

**Non-negative matrix factorization analysis revealed a previously unappreciated transcriptomic regionalization during mucosal healing**. Even though the majority of the tissue was defined by clusters equally represented on both time points, some clusters displayed a partial or drastic enrichment during tissue healing. Cluster 3 (found in the distal colon at the interface between the LP and the muscularis layer), cluster 11 and 16 (localized in the damaged area of d14 distal colon) and cluster 13 (marking lymphoid follicles) were drastically enriched during d14 (Fig. 4a and Supplementary Fig. 5). To visualize how the process of mucosal healing alters the transcriptomic landscape of the colon, we deconvolved the d0 and d14 datasets jointly into 20 factors using NNMF (Supplementary Figs. 6 and 7). Among these, 8 factors were defined by genes expressed in specific regions during mucosal healing (d14), but not at d0 (Fig. 4b). In the proximal colon, factor 1 was characterized by genes involved in bile acid and fatty acid metabolism (e.g. *Cyp2c55*, an enzyme involved in the metabolism of 19-hydroxyeicosatetranoic acid). By contrast, factors positioned in the distal colon were characterized by genes involved in inflammatory processes (e.g. *Duoxa2* and *Il18*) and tissue remodeling (e.g. *Col1a1* and *Col1a2*), among others (Fig. 4c). Due to their close proximity and their marked enrichment during mucosal healing, we focused on factors 5, 7, 14, and 20. Factor 5 delineated an edematous area, which was histologically characterized by inflammation and positioned right beneath a severely injured epithelial layer with complete loss of crypt architecture (i.e. factor 14). Pathway

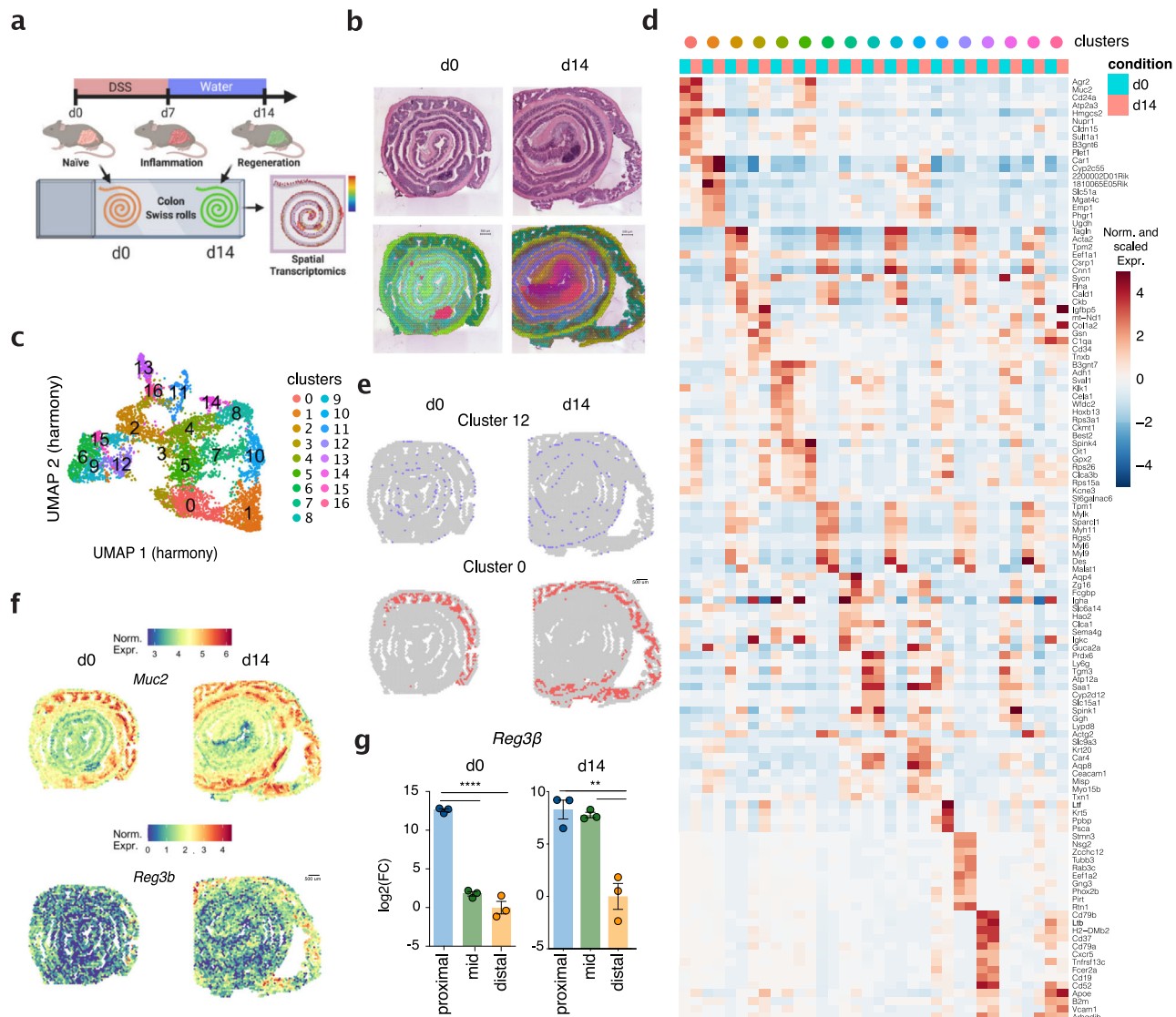

**Fig. 3 Changes of the molecular topography during mucosal healing are dominant at the distal colon. a** Schematic representation of the experiment: colitis was induced by dextran sodium sulfate (DSS) administration in drinking water for 7 days followed by 7 days of regular water to promote tissue repair. Colonic tissue from a wild-type naive mouse (d0, from Fig. 1) and from a mouse undergoing colonic regeneration (d14) were processed as Swiss roll for spatial transcriptomic using Visium 10X technology. (n = 1 Swiss roll per time point). **b** Top: Hematoxylin and eosin staining of colonic tissue from d0 and d14. Bottom: spatial representation of UMAP values in CMYK colors on colon d0 and d14. Spots with the same color in the two time points represent transcriptionally similar regions. **c** Uniform Manifold Approximation and Projection (UMAP) representation of 16 color-coded clusters defining regional transcriptome diversity in the colonic d0 and d14 datasets combined. **d** Heatmap showing expression of top genes defining each cluster (color-coding on top) in the ST datasets from the two timepoints (light blue columns: colon d0; pink columns: colon d14). **e** Schematic representation of cluster 0 and cluster 12 distribution in colon d0 (on the left) and d14 (on the right). **f** Expression of selected genes in cluster 0 onto ST. **g** qPCR validation of regional expression of *Reg3b* in proximal, mid and distal colonic biopsies from wild type mice at steady state conditions (d0) and during mucosal healing (d14) (n = 3, each dot represents one mouse). Data are presented as mean values ± SEM. Significance was assessed by one-way ANOVA with Bonferroni post-test. **p < 0.01; ****p < 0.0001.

analysis revealed that factor 5 was associated with processes involving anatomical structure development, cell adhesion, and extracellular matrix (ECM) organization (Fig. 4d). Among the top genes defining this factor, we found *Igfbp5* and *Igfbp4*, as well as collagens *Col1a1* and *Col1a2* (Fig. 4c). In contrast, factor 14 was characterized by the expression of genes involved in stress response (e.g. *Duoxa2* and *Aldh1a3*) and leukocyte infiltration (e.g. *Ly6a* ansd *Cxcl5*), which indicate an acute response to a barrier breach and tissue damage.

At the end of the colonic tissue, the anus separates a mucosal tissue with a monolayered epithelium (the rectum) from a stratified squamous epithelium (skin). Homeostatic breakdown resulting from colonic inflammation generates an area of epithelial instability wherein a heterogeneous tissue at the interface between skin and colonic epithelium appears (i.e. enlarged multilayered crypt-like structures with squamous, but not cornified, appearance). Factor 7, characterized by the expression of several keratins (e.g. *Krt13*, *Krt5*, *Krt14* and *Krt6a*) and pathways involving keratinocyte differentiation and wound healing, delineated this area (Fig. 4c, d). Finally, factor 20 predominantly defined the distal epithelium undergoing hyperplasia and crypts arborization, which indicates epithelial repair. Gene ontology revealed that this factor was associated with organogenesis (e.g. *Hoxb13*) and response to organic substrates

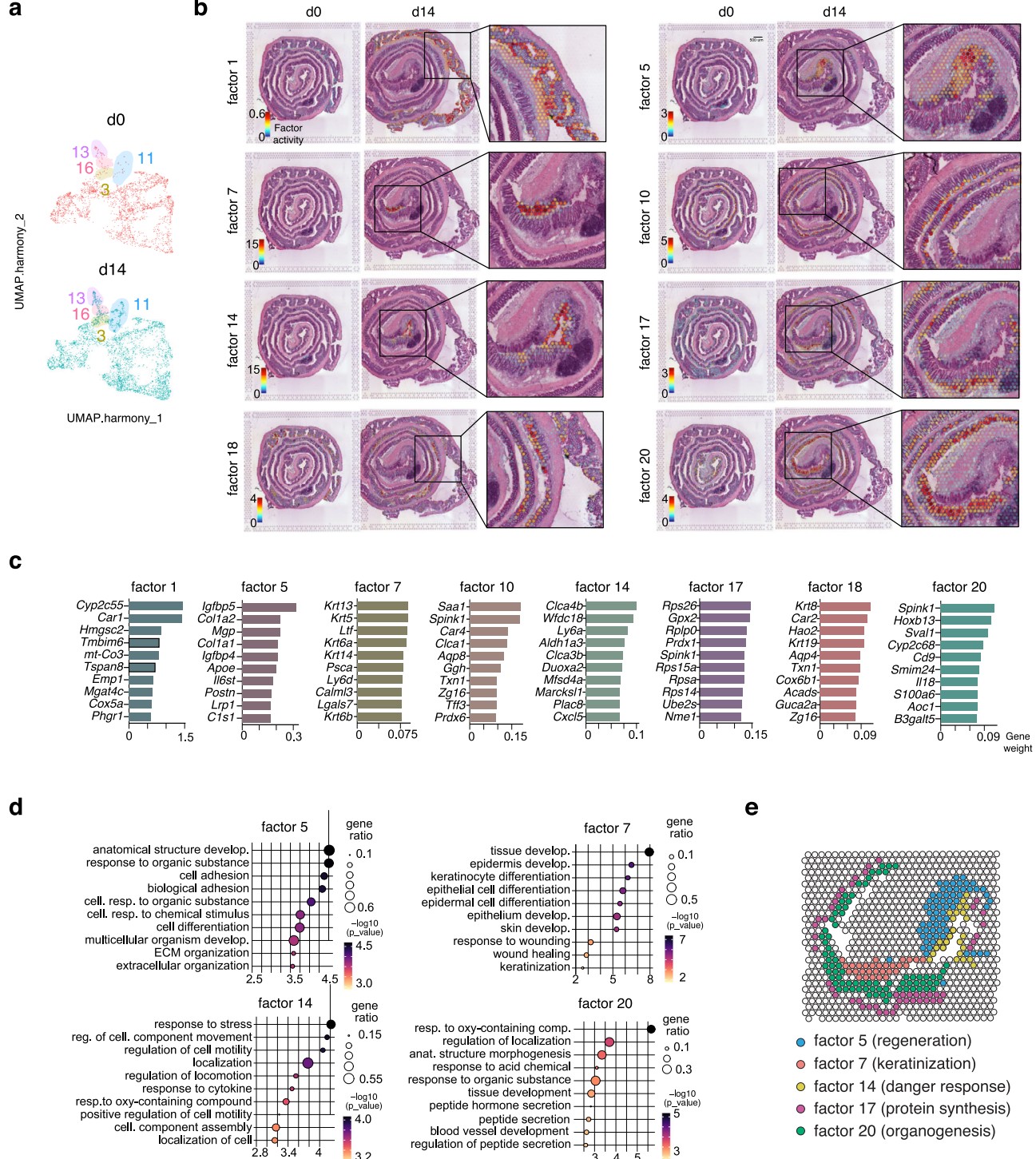

**Fig. 4 Non-negative matrix factorization reveals eight distinct molecular patterns during colon mucosal healing. a** UMAP representation of 16 clusters in d0 and d14 colon. **b** Hematoxylin and eosin images displaying overlaid spots with the highest factor weight. **c** Top 10 genes defining the indicated NNMFs (factors). **d** Functional enrichment analysis (Gene Ontology, GO) based on the top genes defining factor 5, 7, 14 and 20. **e** Schematic representation summarizing the expression pattern between selected factors. Biological processes associated with each factor are indicated in brackets.

(e.g. *Cyp2c68*) (Fig. 4c, d). Analysis of a second d14 dataset (GSE190595) showed similar results as processes associated with "keratinization" (new Factor_11'), "danger response" (new Factor_15'), and "regeneration" (new Factor_17') consistently emerged during mucosal healing (Supplementary Fig. 8).

Overall, DSS-induced injury resulted in an assorted co-occurrence of different histopathological processes within the murine colon. Furthermore, our analysis revealed a previously unappreciated heterogeneous transcriptional and regional landscape of tissue repair (Fig. 4e), which varies in location among biological replicates (Supplementary Fig. 8).

**Predictive algorithms revealed coordinated signaling pathways depending on location.** We interrogated if distinct signaling

pathways could be inferred by the spatially organized transcriptional profiles by using PROGENy[28,29]. Unlike other Gene Set Enrichment tools (as KEGG), PROGENy estimates signaling pathway activities by looking at expression changes of downstream genes in signaling pathways, which provides a more accurate estimation of the activity of the pathway. A score for each of the 14 pathways annotated in PROGENy (i.e. Wnt, VEGF, Trail, TNFα, TGFβ, PI3K, p53, NFkB, MAPK, JAK/STAT, Hypoxia, Estrogen, Androgen and EGFR) was estimated for each ST spot on d0 and d14 slides (see methods). First, we computed a correlation matrix to understand how the spatially organized transcriptional programs identified by NNMF (i.e. factors of Fig. 4) could be explained by signaling pathway activities. We observed two main groups, in which, group 2 pathways (Androgen, JAK-STAT, NFkB, TNFα, p53, Hypoxia and Trail), characteristic of an inflammatory/acute response to damage, were associated with factors comprising damaged distal epithelium (e.g. factor 7, 10, 14, 20) and proximal epithelium (factor 11, 19) (Fig. 5a, b). In contrast, group 1 pathways (TGFβ, Wnt, PI3K, Estrogen and EGFR), normally regulating pro-regenerative/tissue remodeling processes, were associated with factors defining the tissue beneath the damaged epithelium (e.g. factor 5 and 17), the muscle layer (factor 2, 6 and 12) and lymphoid follicles (factor 9) (Fig. 5b). In addition, MAPK and VEGF pathways were associated with similar factors, and as expected, the spatial patterns of MAPK and VEGF pathways activity were comparable (Fig. 5c). Of note, at steady state conditions (d0) MAPK and VEGF pathways were homogeneously active along the mid-distal colon, whereas during mucosal healing, their activation was higher within the damaged/regenerating areas (Fig. 5c, black arrows).

**Shared and complementary pathway activities during mucosal healing.** Comparable TNFα, NFkB, and JAK-STAT pathway activation scores between some factors (e.g. factor 10 and 14) (Fig. 5a) suggest interconnectivity between these inflammatory pathways. To test this possibility we further analyze the spatial pattern of these pathways. In particular, the activities of the TNFα and NFkB pathways were almost identical within the colon, regardless of the time point analyzed (Fig. 5d). Higher TNFα and NFkB activities were appreciated in areas associated with injury and ILFs (Fig. 5d). Of note, in the absence of damage/inflammation (d0), the spatial distribution of TNFα and NFkB showed activity confined to the ILF luminal edge (Fig. 5d, d0), in agreement with previous studies showing that TNF drives ILF organogenesis[30]. Whether ILF forms where subclinical local damage occurs or whether their presence, which allows dynamic exchange with the external environment, causes subclinical inflammation, remains to be explored.

In contrast, JAK-STAT pathway activation showed co-occurrence with TNF and NFkB mostly in the damaged area (factor 14), but not in ILFs (Fig. 5d). These results suggest that although all three pathways may play a role within the damaged tissue, TNFα and NFkB, but not JAK-STAT, are involved in the formation/function of ILFs. Unlike these pathways, androgen and estrogen pathways showed mutually exclusive patterns of activity. Higher androgen pathway activity was observed in areas of injured epithelium, while higher estrogen activity was associated with the muscle layer (arrows, Fig. 5e), suggesting that these pathways negatively regulate each other during mucosal healing.

**Low p53 pathway activity is associated with proliferating crypts.** Activation of the p53 pathway was homogeneously distributed across the proximal-distal axis, but it showed more activity in the luminal side compared with the LP and muscle layer (Fig. 5f). Interestingly, p53 activity was lower in the damaged area (box ii in Fig. 5f). Activation of p53 triggers cell

cycle arrest, senescence, and apoptosis[31], suggesting that spots with decreased p53 activity might be enriched in proliferating cells within the damaged area. To test this possibility, we overimposed lower p53 activity spots onto the H&E images and showed co-localization with the bottom of crypts (Fig. 5f, H, E boxes). To investigate if proliferating stem cell signatures co-localize with low p53 activity spots, we took advantage of our single cell RNA sequencing (scRNAseq, GSE163638) dataset of intestinal epithelial cells from d14 colon and identified a population of proliferating stem cells (Fig. 5g). We mapped the stem cell core signature onto the d14 colon ST datasets, and we superimposed the spots with high scores in the stem cell core onto the H&E section. In agreement with our hypothesis, spots containing high scores (Fig. 5g) coincided with low p53 activity (Fig. 5f). Pearson correlation analysis confirmed that ST spots with high stem cell scores negatively correlated with p53 activity (Fig. 5h). Thus, our data suggest that low p53 activity allows the identification of proliferating crypts during mucosal healing. In summary, we spatially positioned clinically relevant pathways predicted by PROGENy and showed that these pathways are highly coordinated during mucosal healing.

**Integration of human datasets with mouse spatial transcriptomic.** To enquire about the translational potential of the murine colonic ST, we investigated whether human datasets could be integrated into murine ST data. Towards this end, we took advantage of a human developing gut dataset[32] and mapped 31 distinct epithelial and stromal cells onto our ST dataset (Fig. 6a). We observed correlations between human cell types and distinct murine ST factors (Fig. 6b), indicating specific localization of human cell signatures within the mouse colon. Interestingly, the signature of human proximal enterocytes is highly correlated with factor 1 (Fig. 6b), defining the most proximal epithelium in mice (Supplementary Fig. 9a). Human distal enterocytes and absorptive cells highly correlated with factors 3 and 10 (Fig. 6b), which defines the most distal epithelium in mice (Supplementary Fig. 9a). These results indicate that the transcriptomic features defining proximal and distal epithelial cells are conserved between mouse and humans. On the other hand, two human stromal cells characterized by the expression of the chemokines CCL21 and CXCL13 uniquely and strongly correlated with factor 9, defining lymphoid follicles (Fig. 6b), which is in agreement with the well-known role of these chemokines in ILF development[33]. Interestingly, these stromal cells mapped in complementary patterns within the mouse ILF (Fig. 6c), suggesting that the coordinated action of these cells might determine the recruitment/localization of immune cells within the follicle. Next, we analyzed the damage/regeneration area (factor 5 and 14) which correlated with S1 (Stromal 1, fibroblast marking bulk of submucosal structural cells in human), S1-COL6A5 and S1-IFIT3 human cells (two subtypes of S1) (Fig. 6b) and mapped in a complementary pattern of distribution (Supplementary Fig. 9b).

We then extend our analysis to other cell types during mucosal healing (Fig. 6d and Supplementary Fig. 9c). Interestingly, immune cells, mesothelium, endothelium and fibroblast signatures were spatially enriched within defined areas during mucosal healing. In particular, immune cells were associated with factor 9 (lymphoid follicles), whereas mesothelial cells localized within factor 7 (keratinization) (Fig. 6d and Supplementary Fig. 9d). Among 11 distinct immune cell types identified[32], monocytes and SPP1 + macrophages were enriched in factor 14 (danger response) and in factor 5 (tissue remodeling) respectively, in line with their known roles in acute response to injury and matrix deposition/ wound healing (Supplementary Fig. 9e). In contrast, lymphocytes were enriched in factor 9 (lymphoid follicles) (Fig. 6e and

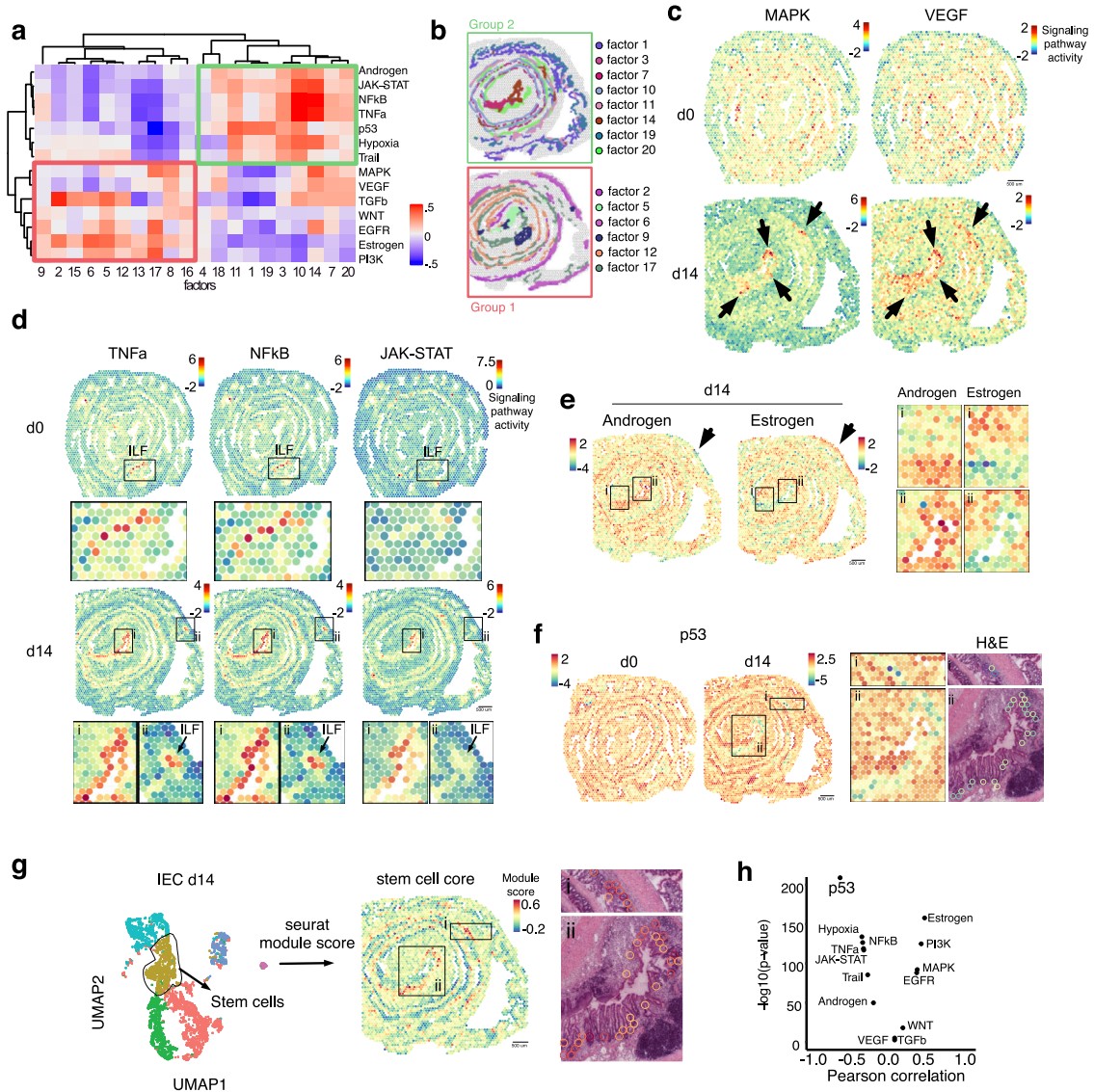

**Fig. 5 Predictive algorithm reveals pathway-specific spatial patterns during mucosal healing. a** Correlation matrix between non-negative matrix factorization (NNMF) and pathway activity scores determined by PROGENy. **b** Schematic of the colon area at d14 displaying the distribution of the indicated factors. **c** Spatial transcriptomic (ST) spot heatmaps of the colon at d0 (upper swiss rolls) and d14 (lower swiss rolls) showing pathways scores predicted by PROGENy. Arrows indicate areas of tissue damaged as defined by factor 14 shown in Supplementary Fig. 7 and Fig. 4a–c. **d** ST spot heatmaps showing TNFα, NFkB, JAK-STAT pathway activity on d0 (upper Swiss rolls) and d14 (lower Swiss rolls). Selected areas indicated as ILF (isolated lymphoid follicles) or "i" and "ii" and outlined in black are magnified below each Swiss roll. Arrows in "ii" indicate the presence of an ILF. **e** Spatial distribution of androgen and estrogen pathway activity at d14. Selected areas (indicated as "i" and "ii") are magnified. Arrows indicate an example of the muscle layer showing opposite expression patterns between the two pathways. **f** Spatial distribution of p53 pathway activity at d0 and d14. Selected areas indicated as "i" and "ii" on colon d14 are magnified on the right. Hematoxylin and eosin magnifications show the overlaid spots with the lowest p53 activity shown in "i" and "ii". **g** Left: UMAP visualization of intestinal epithelial cells (IEC) clusters from scRNAseq on colon d14 (GSE163638). Middle: ST spots from colon d14 are color-coded based on the enrichment of stem cell core signature identified from scRNAseq dataset. Selected areas indicated as "i" and "ii" on colon d14 are magnified on the right. Right: Hematoxylin and eosin magnifications showing the overlaid spots with the highest stem cell signature shown in "i" and "ii". **h** Pearson correlation between PROGENy predicted pathways and stem cell signature on the ST dataset.

Supplementary Fig. 9e). Further analysis showed how these immune cells are heterogeneously distributed within the ILF (Fig. 6e). These results provide a proof-of-concept and support the notion that principles of spatial distribution within the colonic tissue appear to be conserved between species and highlight murine ST as a valuable platform for exploring and translating findings on distribution patterns of cells/genes within a tissue.

**Mapping transcriptomic datasets onto ST to interrogate temporal dynamics of tissue healing**. In order to establish a

framework to integrate existing knowledge with ST datasets, we took advantage of our longitudinal RNAseq dataset of colonic tissue collected during acute epithelial injury and the recovery phase in the DSS-induced colitis model[10]. In this study, we identified sets of genes (called modules) displaying characteristic expression patterns, with some genes being: a) downregulated upon injury (modules 2, 8, 7), b) upregulated during the acute/inflammatory phase (modules 1, 3, 4, 9), and c) upregulated during the recovery phase of DSS colitis (modules 5, 6)[10] (Fig. 7a). To identify whether the different temporally regulated processes (i.e. modules) were enriched in specific areas of the tissue, we

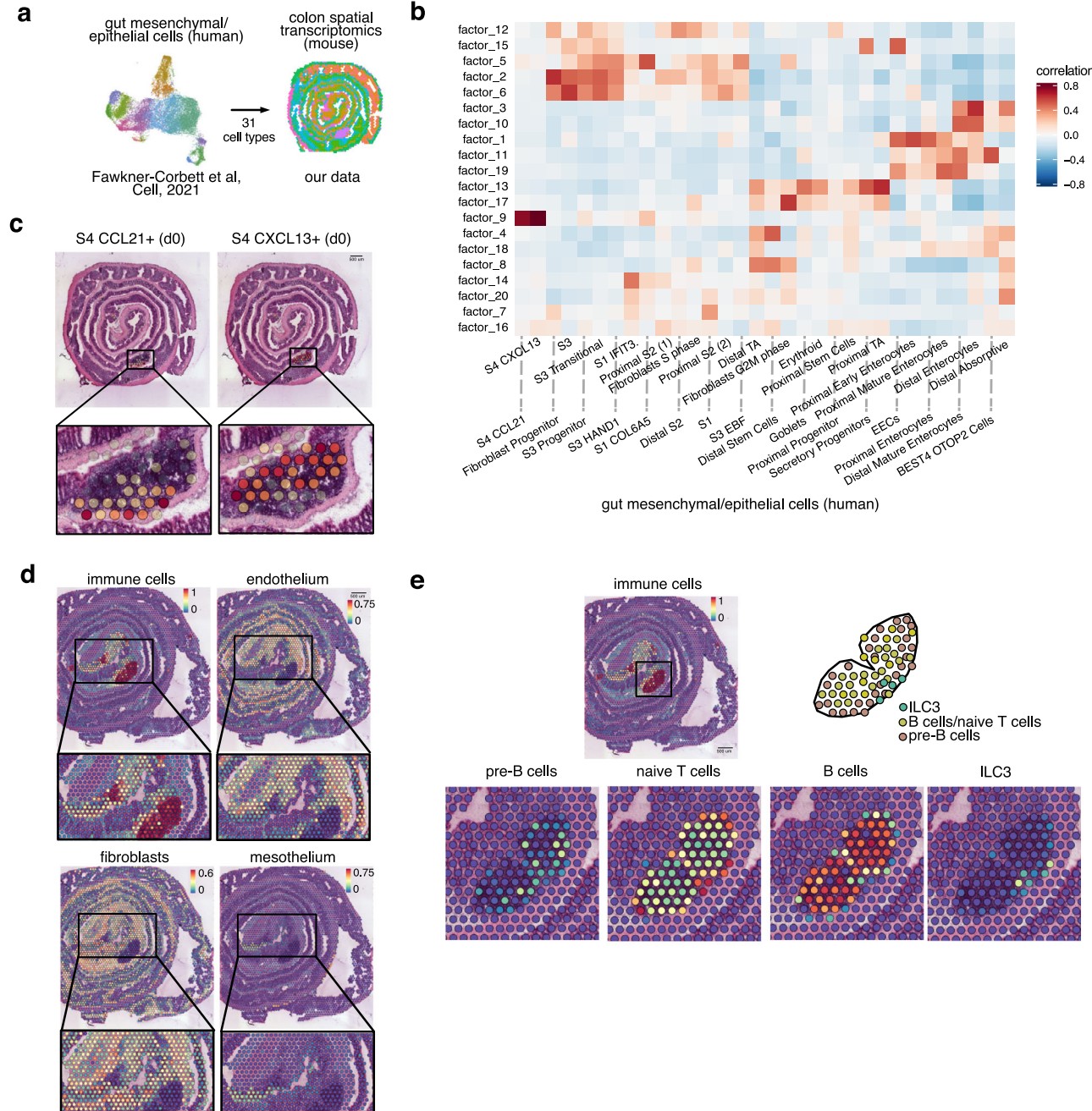

**Fig. 6 Human cell type mapping onto murine spatial transcriptomic datasets. a** Scheme showing the integration of published human single cell RNAseq[32] and our mouse Visium datasets. **b** Correlation matrix between transcriptomic profiles from human single cell datasets[32] and factors defining transcriptomics patterns in mouse ST. **c** Integration of human stromal cell transcriptomic profiles (S4.CCL21+ and S4.CXCL13 + ) onto visium datasets at day 0. **d** Integration of human intestinal cell transcriptomic profiles onto visium datasets at day 14. **e** Integration of human immune cell transcriptomic profiles onto visium datasets at day 14 and magnification of the isolated lymphoid follicle area.

computed a correlation matrix between the gene signature of modules and ST factors (Fig. 7b). Higher gene enrichment was found in module (m)1 and m6, characterized by genes induced during the inflammatory and recovery phase, respectively (Fig. 7b). Among the genes shared between factor 9 and m1, *Ptprc, Cd72* and *Lyz2* encode for proteins expressed by immune cells and map predominantly to lymphoid follicles and damaged areas in d14 (Fig. 7c"i"). In contrast, ~60% of the top driving genes defining factor 15 (i.e. ENS) were shared with m6, and their expression was distributed in the submucosa/muscularis layer where neuronal bodies reside (Fig. 7d"i"). GO enrichment analysis

of m1 and m2 also confirmed that the most dominant pathways were involved with inflammatory responses and chemical synaptic transmission (Fig. 7c"ii", d"ii"). The correlation between temporal transcriptomic modules and spatial factors suggests that factor 9 (lymphoid follicles) and factor 15 (ENS) are characterized by an ongoing inflammatory and regenerative profile, respectively. Thus, the integration of longitudinal and ST data can be a powerful tool to unveil biological processes related to diseases in time and space.

**Spatial distribution of genes defining UC1 and UC2 profiles.** We then sought to investigate if clinically relevant patient gene

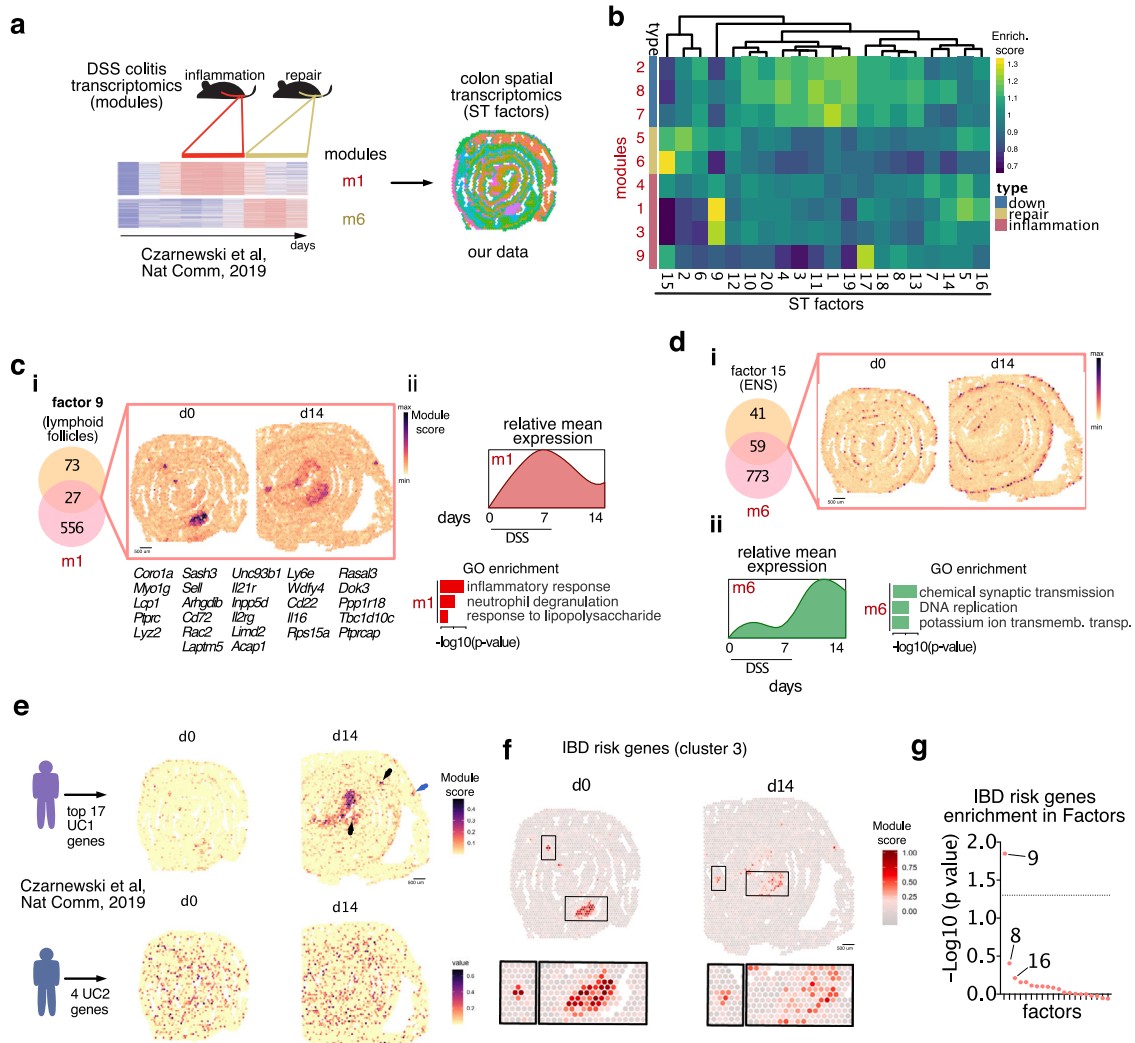

**Fig. 7 Spatial transcriptomic (ST) allows mapping transcriptomic signatures with clinical relevance. a** Scheme illustrating the analysis designed and dataset sources. **b** Correlation matrix between transcriptomic modules distinguishing the processes of inflammation and mucosal healing during DSS-induced colitis[10] and factors defining transcriptomics patterns in ST. **c** (i) Venn diagram and spatial representation of overlapping genes between module 1 and factor 9, (ii) relative mean expression and Gene Ontology (GO) of genes belonging to module 1. **d** (i) Venn diagram and spatial representation of overlapping genes between module 6 and factor 15, (ii) relative mean expression and Gene Ontology (GO) of genes belonging to module 6. ENS: enteric nervous system. **e** Spatial distribution of genes defining UC1 and UC2 patients on mouse ST colon d0 and d14. UC: Ulcerative Colitis. **f** Spatial distribution of Inflammatory Bowel Disease (IBD) risk genes from Cluster 3 (Supplementary Fig. 10b) on colon d0 (left) and d14 (right). **g** Gene Set Enrichment Analysis for IBD risk genes in Cluster 3 (Supplementary Fig. 10b) and NNMF factors (Supplementary Figs. 6 and 7).

signatures could be mapped onto mouse ST datasets. Toward this end, we used the recent gene signature identifying ulcerative colitis (UC) subgroups of patients: UC1 and UC2[10]. In this previous study[10], by integrating human IBD and murine DSS transcriptomic datasets, we identified UC1 and UC2 as two subgroups of IBD patients characterized by comparable histological disease severity but different underlying genetic signature. This molecular classification is clinically relevant because the UC1-related transcriptomic signature is associated with poor responses to biological therapies and is enriched with genes involved in neutrophil activity. In contrast, approximately 70% of UC2 patients achieved a clinical response to *anti-TNF* antibodies[10]. In addition, Smillie et al.[34], showed that inflammatory fibroblasts and monocytes mainly drive anti-TNF resistance, and many of the genes upregulated in UC1 patients' colonic tissue are highly expressed by these cell types. To further understand the spatial distribution of differentially expressed genes between UC1 and UC2 patients, we mapped all genes

upregulated in either UC1 and UC2 patients onto colon ST. Whereas genes defining UC2 patients were homogeneously expressed across the colon (d0 and d14) (Fig. 7e), genes defining UC1 were mostly localized within the damage/repair area at d14 (Fig. 7e). Further investigation revealed that all the main functional classes upregulated in UC1 patients, including collagen synthesis (*Col12a1*, *Col4a1*, *Col4a2*, *Col7a1*), ECM breakdown (*Mmp3*), Wnt-signaling pathway (*Wnt5a*), cytokine signaling (*Il11*, *Il1b*, *Il33*, *Il1r2*, *Il1rn*, *Tnfrsf11b*, *Csf2rb*, *Csf3r*, *Socs3*, *Trem1*, *Cxcr2*) and innate immunity (*S100a9*, *S100a8*, *C5ar1*, *Sell*, *S100a4*), were also upregulated in areas of tissue damage (Supplementary Fig. 10a). In summary, these results suggest that UC1 patients may possess higher tissue damage and ulceration compared with UC2 patients.

**Defining the topography of IBD risk genes.** Spatial transcriptomics of the colon undergoing injury/repair provides an opportunity to comprehensively map IBD-associated risk genes.

Therefore, we interrogated the expression pattern of various human IBD-risk genes[35–39] on the ST profile of d0 and d14 colonic tissue. Out of the 122 interrogated genes, 95 IBD-risk genes were selected based on the existence of their murine ortholog and their detectable expression in the ST dataset. In order to identify whether the spatial expression of these variants defined topographic patterns within the tissue, we computed a correlation matrix (Supplementary Fig. 10b). Cluster analysis resulted in three main co-expression clusters, with cluster 3 possessing the highest spatial expression correlation between genes (Supplementary Fig. 10b). Functional annotation of the genes within this cluster revealed enrichment in pathways related to immune cell recruitment (e.g. *Itgal, Icam1, Itga4*), activation (e.g. *Cd6, Plcg2, Ncf4, Il10ra*), and antigen presentation (e.g. *Tap1, Tap2, Psmb8*). To understand the spatial distribution of genes belonging to cluster 3, we mapped them onto the colonic tissue using ModuleScore, a Seurat function assigning a score in the ST dataset to a set of predefined genes (i.e. cluster 3 genes). In line with the functional annotation of cluster 3 genes, we observed enrichment in lymphoid follicles areas both on d0 and d14 (Fig. 7f). To understand which ST factors were enriched with IBD risk genes, we performed Gene Set Enrichment Analysis (GSEA) and calculated overrepresentation scores of IBD-risk variants in the NNMF dataset (Supplementary Figs. 6, 7). This analysis showed that factor 9, defining lymphoid follicles within the tissue (Supplementary Fig. 6), was the only factor with significant enrichment of IBD risk genes (Fig. 7g). Altogether, this analysis revealed that the expression of a subset of human IBD-risk genes spatially co-occur within the murine colon. Their specific expression pattern suggests that colonic tissue lymphoid follicles might define the area to potentially target when developing therapeutic strategies for IBD patients displaying aberrant immune activation.

## Discussion

We and others have deeply characterized the transcriptomic landscape during mucosal healing in the colon and small bowel[10,40,41]. However, these studies lacked the spatial resolution describing where genes were expressed. Here, we spatially placed cell populations and pathways that might play pivotal roles in driving tissue response to damage. The current study uncovered spatial transcriptomic patterns that are present at steady state conditions and that arise in response to damage; these spatial transcriptomic patterns were characterized by unique transcriptional signatures and coincided with different histological processes. Moreover, we profiled the regional distribution of different biological processes, such as acute response to injury or a regenerative response. Finally, we demonstrated the translational potential of this dataset as seen by conserved spatial localization of gene signatures in human tissue and transcriptomics data and by testing the distribution of clinically-relevant genes.

A recent study characterized the transcriptomic landscape during human gut development[32]. Here, we further these results by taking advantage of murine colonic Swiss rolls that fit the 6.5 mm² area constraints provided by the manufacturer to perform spatial transcriptomics. This approach enabled us to visualize the transcriptomic landscape of the whole colon in the same slide, including the most proximal and distal segments. Using bioinformatics tools (NNMF analysis), we uncovered a previously unappreciated molecular regionalization of the colonic tissue in steady state conditions. This analysis allowed the identification of distinct epithelial, LP, and muscularis/submucosa genetic programs depending on their proximal to distal colon localization. Importantly, when mapping human cells into our ST datasets, we observed conservation in transcriptomic features defining proximal and distal locations, suggesting that our newly described molecular segmentation is conserved across mammals.

Using the entire murine colon, we provided a detailed analysis of a previously unappreciated compartmentalization of the tissue repair process. In line with previous reports[42], our unbiased analysis of the transcriptomic landscape during mucosal healing reveals that while dramatic transcriptomic changes occur in the distal colon, the proximal colon remains almost comparable to the steady state. Two potential scenarios can be proposed: (a) the level of damage is homogenous and the proximal colon heals faster compared with the distal colon; or (b) the proximal colon is more protected compared with the distal colon. Dramatic changes in the distal rather than the proximal colon are in agreement with the phenotype observed in UC patients, where the focus of inflammation extends proximally from the rectum[43]. In addition, genes and/or pathways, such as the JAK-STAT and TNFα pathway[44] or genes/pathways characterizing UC1 patients[10], were found to be dominant in the distal colon, suggesting that DSS-induced colitis is a clinically relevant experimental model of UC1. Whether higher levels of damage/tissue repair in the distal colon depend on microbiota, host-induced responses to the microenvironment, or just different kinetics, remains to be addressed.

At steady state conditions, we identified molecular signatures associated with lymphoid structures. Unlike Peyer's patches (PP) that are macroscopically visible in the murine small intestine, CP and ILF cannot be dissected and analyzed separately from the rest of the colonic tissue for transcriptomic readouts. For instance, enrichment in the NFkB and TNFα pathway activity was detected in the lumen-facing area corresponding to the epithelial layer overlaying lymphoid clusters in steady state. Because these pathways are usually associated with immune activation/inflammatory responses, these data suggest that ILF-associated epithelium is undergoing inflammation. Moreover, expression of clusterin (*Clu*) alone was found to be highly specific and sufficient in defining the isolated lymphoid follicles (ILF). Previous studies have reported Clu expression in follicular dendritic cells (FDC)[45], as well as M cells[21] in the Peyer's patches. While *Clu* expression in FDCs serves as a pro-survival factor for germinal center B cells in the follicle, the role of Clu in M cells is not clear. Interestingly, a recent study[40] identified *Clu* as a marker of intestinal stem cells (ISC), known as revival stem cells, which are rarely found in steady state, but are predominantly found in regenerating intestine following injury. Whether the expression of *Clu* in the ILF- and follicle-associated epithelium has any bearing during colonic infection and regeneration needs to be further investigated.

Besides the proximal-distal variance in the transcriptomic alterations during tissue repair, our NNMF analysis revealed a high degree of compartmentalization within the distal colon itself. At least 5 factors were delineating topographically and transcriptionally distinct areas in the distal colon undergoing different biological responses to tissue injury. Such heterogeneity is likely the result of different healing programs, such as skin-like re-epithelialization (factor 7), acute damage (factor 14) and tissue regeneration (factor 20). Supporting this notion, the integration of our RNAseq kinetic dataset[10] into the ST map revealed that certain areas of the distal colon were transcriptionally closer to samples from the acute phase of DSS colitis (i.e. d6 to d8). Whether the spatial transcriptomic signature during the acute phase (i.e. DSS d7) would be associated with an expansion of the damage-associated factors identified at d14 remains to be addressed. Nonetheless, the asynchronous nature of the healing process observed at d14 may be associated with varying degrees of exposure to the external environment and the elaborate architecture of the tissue. Similarly, in human IBD, the "patchiness of the inflammatory response" is a well-known characteristic of Crohn's disease. In UC, the inflammation is traditionally thought

to be continuous with increasing intensity in distal colon, but longitudinal sampling has revealed episodes of both macroscopic and microscopic patchiness of inflammation[46]. Our dataset thus provides a valuable resource to interrogate the transcriptional programs underlying distinct temporal and biological processes of tissue healing.

PROGENy allows the prediction of pathways activated in specific regions of the colon. Our analysis revealed a strong correlation between pathway activities, such as TNFα and NFkB, suggesting that one pathway might depend completely on the other. We also identified that the area of injury is characterized by several pathways that are also increased on the ILF edge at steady state conditions, suggesting that the formation of lymphoid follicles result in the induction of damage-associated pathways. Finally, decreased p53 activity represents a good strategy to identify damage-associated proliferating crypts. These data suggest that within the same region, intestinal crypts are heterogeneous in their response to damage; our dataset provides a toolkit to investigate this composite response.

Among limitations when using spatial transcriptomics technology is that each observation (spot) typically represents the averaged expression profile from multiple cells. For the 10x Visium technology, the diameter of a spot is 55 microns, meaning that mixtures of 1-10 cells are profiled simultaneously. Here, we demonstrate how these mixed profiles can be deconvolved by NNMF to identify transcriptional programs in a broader tissue context. The characteristics of these transcriptional programs are further supported by morphological detail at single- cell resolution obtained from HE bright field images. However, we anticipate that integration with complementary technologies with higher resolution, such as scRNA-seq, would deepen our understanding of the cellular networks and communication pathways underlying these transcriptional programs. Although ST is currently limited to the analysis of mRNA, we anticipate that multimodal spatial profiling will help pinpoint pivotal events in tissue regeneration. For example, parallel targeting of proteins would not only provide a more comprehensive molecular map of the colon, but be particularly useful to identify cell-cell crosstalk events identified by ligand-receptor profiling. This information could help us elucidate crosstalk between epithelial and immune cells which have been implicated in the orchestration of the wound healing process.

Our study provides evidence that ST can be used to map clinically relevant genes and pathways. Genes characterizing a newly described UC subgroup were associated with poor treatment response in the damaged area of the regenerating colon. Previous studies confirm these results; increased inflammation severity predicted poor response to anti-TNF treatments[47] or blocking cell recruitment to the inflamed intestine using anti-a4b7 antibodies[48]. As a result of severe inflammation, colonic ulceration may lead to lower therapeutic responses due to decreased blood drug concentration/drug leakage[49]. Our data suggest that UC1 patients have increased tissue damage compared with UC2 patients, which might contribute to their poor response to biological therapies.

## Methods

**Mice**. Female wild type C57BL/6 J mice between 8 and 10 weeks of age were purchased from TACONIC. Mice were maintained under specific pathogen-free conditions at Karolinska Institutet. All mice were housed in colony cages in a pathogen-free environment with the temperature maintained at 21–23 °C and relative humidity at 50–60%, and were under a 12 h light/12 h dark cycle. All mice were fed ad libitum with standard chow diet.

One female mouse was used to generate the d0 ST dataset and two female mice were used to generate the d14 datasets. Three mice were used to perform the qPCR validation data in Figs. 1g, 3g. All experimental procedures were performed according to national (Sweden) and institutional (Karolinska Institutet) regulations

and guidelines. Animals were maintained under specific pathogen-free conditions at AKM animal Facility (Stockholm, Sweden) and handled according to protocols approved by the Stockholm Regional Ethics Committee (ethical permit number: AKM 3227-2017).

**DSS colitis model**. Mouse colitis was induced by administration of 2% w/v of dextran sodium sulfate (DSS, TdB Consultancy, MW = 40 kDa) dissolved in drinking water ad libitum for 7 days, followed by 7 days of regular water. General health and body weight were monitored regularly. On day 14, colons from untreated and DSS-treated mice were harvested, measured, and processed for spatial transcriptomic studies.

**Tissue processing and spatial transcriptomics**. Colonic tissues from untreated and DSS-treated mice were cleaned from adipose tissue, and cut longitudinally; the luminal content was removed by washing it in cold phosphate buffered saline (PBS). Starting from the most distal portion (i.e. rectum) and with the luminal side facing upward, the colon was rolled resulting in a Swiss-roll with the distal colon in the center and the proximal colon in the outer portion of the roll. The Swiss roll was placed in a histology plastic cassette and snap frozen for 1 min in a bath of liquid nitrogen-cooled isopentane. The frozen tissue was then embedded in Optimal Cutting Temperature compound (OCT, Sakura Tissue-TEK) on dry ice and stored at −80 °C. OCT blocks were cut with a pre-cooled cryostat at 10um thickness, and sections were transferred to fit the 6.5mm² oligo-barcoded capture areas on the Visium 10x Genomics slide. Before performing the complete protocol, Visium Spatial Tissue Optimization (10x Genomics) was performed according to manufacturer's instructions, and the fluorescent footprint was imaged using a Metafer Slide Scanning Platform (Metasystems). Thirty minutes was selected as optimal permeabilization time.

The experimental slide with colonic tissue from d0 and d14 was fixed and stained with hematoxylin and eosin (H&E) and imaged using a Leica DM5500 B microscope (Leica Microsystems) at 5X magnification. The Leica Application Suite X (LAS X) was used to acquire tile scans of the entire array and merge images. Sequence libraries were then processed according to manufacturer's instructions (10x Genomics, Visium Spatial Transcriptomic). After the second cDNA strand synthesis, cDNA was quantified with quantitative RT-PCR ABI 7500 Fast Real-Time PCR System and analyzed with ABI 7500 Software 2.3.

**Quantitative PCRs**. Tissue biopsies (0.5–1 cm long) from wild type C57BL/6 mice were collected from the proximal (right after the cecum), mid (middle portion of the colonic tissue) and distal (last portion of the rectum before the anus) parts of the colon. The biopsies were stored in RNAlater (Invitrogen) at −20 °C. To extract RNA, tissues were placed in RLT Plus Buffer (Qiagen) with 2% b-mercapto-ethanol and lysed by bead-beating (Precellys). RNA isolation was performed using the RNAeasy Mini KIt (Qiagen) according to manufacturer's instructions and was quantified by NanoDrop. RNA was reverse transcribed using iScript RT Supermix (Biorad) and quantitative real time PCR was performed using iTaq Universal SYBR Green Supermix (Biorad) according to manufacturer's instructions. Log2 fold change (FC) was calculated relative to /Hprt using the 2-(DDCT) method. Primers sequences are indicated below:

*Hprt*: TCAGTCAACGGGGGACATAAAGGGGCTGTACTGCTTAACCAG
*Hmgcs2*: AAGGATGCTTCCCCAGGTTCCCAGGTGGAGAAGTTCACC
*Ang4*: CAGCTTTGGAATCACTGTTGGAAGCATCATAGTGCTGACGTAGG
*B4galt1*: ATCAGGCTGGAGACACCAT TGAGAGCAGAGACACCTCCA

**Immunohistochemistry from Human Protein Atlas**. To validate selected genes, the Human Protein Atlas (https://www.proteinatlas.org/), a publicly available repository of validated immunohistochemical stainings of human tissue, was used. Protein expression is illustrated in colonic samples from healthy donors.

**Sequencing and data processing**. Visium libraries were sequenced on a NovaSeq S1 flow cell (Illumina), at a depth of 196-259 million reads per sample, with 28 bases from read 1 and 120 bases from read 2. Demultiplexed fastq files (read 2) were trimmed to remove polyA homopolymers and TSO adapters using Cutadapt[50]. Briefly, either partial of full-length TSO adapter sequences were removed from the 5' end of read 2 by setting the TSO sequence (AAGCAGTGGTATCAACGCAGAGTACATGGG) as a non-internal 5' adapter (minimum overlap set to 5 and error tolerance set to 0.1). PolyA homopolymer sequences were removed by setting a sequence of 10 as a regular 3' adapter (minimum overlap set to 5). Raw read 1 and trimmed read 2 fastq files were processed with the spaceranger command line tool (version 1.0.0, 10X Genomics) and mapped to the pre-built mm10 reference genome (GRCm38, patch release 6).

**Filtering and normalization of colon Visium data**. Processed gene expression matrices for the d0 and d14 tissue sections were merged and converted into a Seurat object using InputFromTable (STutility). The merged dataset was then enriched for protein coding genes by removing genes annotated with a non coding RNA biotype. The filtered dataset was then normalized by variance stabilizing transformation using SCTransform, setting genes with a residual variance higher

than 1.1 as variable features (return.only.var.genes = FALSE, variable.features.n = NULL, variable.features.rv.th = 1.1).

**Deconvolution of colon transcriptomic data.** Using non-negative matrix factorization (NNMF), the spatially resolved transcriptomic data set was deconvolved in three different modes. First, the data was deconvolved into 3 factors to capture basic molecular structures of the data. The matrix factorization was conducted using RunNMF (nfactors = 3) on the scaled and normalized expression data, including all variable genes while omitting ribosomal protein coding genes.

For the second mode, the data was first split into d0 and d14 followed by normalization (SCTransform) and deconvolution into 20 factors. Lastly, for the third mode, we deconvolved the merged and normalized expression data (including both d0 and d14) into 20 factors. Top contributing genes were selected from the feature loadings matrix representing the weighted contribution of each gene to the factors.

**Conversion to linear coordinate system.** Images of HE-stained day 0 colon tissue section were first downscaled to $600 \times 541$ pixels. Next, the outer edge of the *muscularis externa* was manually outlined using a 1 pixel diameter brush in Photoshop and defined as the base layer, that is, the base of the y-axis in the new coordinate system. All subsequent steps were computed in R (v.4.0.0). The outlined base layer was imported into R and converted into an unordered set of points (pixel coordinates), each point defining a position on the base layer. In order to define the new coordinate system we aimed to (1) define the endpoints of the base layer and (2) order the points of the base layer from distal to proximal. To solve this problem, a kd-tree based k Nearest Neighbor algorithm (kNN, *dbscan* R package) was used to construct an adjacency matrix with k set to 5. One attribute that is unique to the end points is that they only have neighbors in one direction along the base layer. Therefore, the end points could be defined by searching for the 2 points with the smallest average distance to its 5 nearest neighbors. Next, the knn adjacency matrix was converted into an undirected, connected graph (graph_from_adjacency_matrix, *igraph* R package). From this graph, the shortest path (geodesic) from endpoint to endpoint was detected, that is, the path with the minimal number of vertices (shortest_paths, *igraph* R package). The ordered subset of vertices from the shortest path was then used to order the base layer point set from the proximal to the distal end.

Visium spots were assigned to base layer pixel coordinates using the following steps. First, to limit the search space for each base layer pixel coordinate, only spots within an 80-pixel radius were considered. The search space was further limited by selecting spots with an angle to the tangent to the curve at the base layer point ranging between 35°-125°. The result of this iterative search was a set of base layer points and neighboring Visium spot pairs. To ensure that each Visium spot only paired with one base layer point, only the pair with the smallest Euclidean distance were kept. Finally, x-coordinates of Visium spots in the unrolled coordinate system were defined as the order of the base layer points and the y-coordinate was defined as the distance between Visium spots to the closest base layer point.

**Functional enrichment analysis of factors.** Functional enrichment analysis was conducted on the genes with the highest weighted contribution to each factor. For each factor, the feature loading values lower than 0 were removed and the remaining values were log-transformed and smoothed using gaussian smoothing with a window length of 10. The top contributing genes were then selected using the Unit Invariant Knee (UIK) method based on the sigmoidal shaped curve defined by the smoothed values and their rank (uik function from the inflection R package).

Top contributing genes selected by the UIK method were subjected to functional enrichment analysis using the gprofiler2 R package (organism = "mmusculus", sources = "GO:BP"). The top 20 most significant terms were included in the figures.

**Data integration, and clustering and DEA.** The spatially resolved transcriptomics data from d0 and d14 were integrated across days using harmony[27] (reduction = "pca", assay.use = "SCT"). Unsupervised clustering was performed on the harmony embedding (dims = 1:20) using a shared nearest neighbor approach (FindNeighbors and FindClusters from Seurat) with the resolution set to 0.8. The data was embedded into a 2D UMAP using the first 20 harmony vectors as input to RunUMAP from Seurat. Differential expression analysis (DEA) was conducted using FindAllMarkers from Seurat to detect marker genes for each cluster.

**Estimation of signaling pathway activities in Spatial Transcriptomics.** For each slide, pathway activities of each spot were estimated with PROGENy[28] using the top 1,000 genes of each transcriptional footprint. Input expression matrices of each slide were normalized with *sctransform* implemented in Seurat 3.1.4.9[51]. To further annotate the factors identified in the NNMF of the combined slides, we calculated the Pearson correlation of the spot-level scores of each individual factor with their corresponding spot-level pathway activity in each slide separately.

**Integration of scRNAseq data with ST data set.** To map the gene expression signature of proliferating stem cells from our independent dataset of intestinal epithelial cells on the d14 colonic tissue, we calculated module scores as implemented in Seurat's function *AddModuleScore*[52,53] Next, we calculated the Pearson correlation between each PROGENy pathway score and the stem cell score. P-values were corrected using a Benjamini-Hochberg procedure. The genes used to build the signature were the following: *Hmgb2, Ube2c, Pclaf, Stmn1, Top2a, Tubb5, Birc5, Mki67, Cenpf, Tuba1b, Cenpa, Ccdc34, Tmpo, Cdca3, Ccna2, Cdk1, Nucks1, Smc4, Spc24, Cdca8, Nusap1, Racgap1, Pbk, Kif15, and Mad2l1.*

**Integration of Visium data with human scRNA-seq data.** Single-cell RNA-seq data for cell type populations of the developing intestine were obtained from Fawkner-Corbett et al.[32]. Human gene symbols were converted to their mouse homologs (MGI symbols) using the getLDS function from the *biomaRt* R package. For genes where the mouse homolog was missing, the human gene symbol was kept. Three rounds of label transfer were conducted. The first and second run was conducted using cell type labels defined by Fawkner-Corbett et al., (e.g. S1 IFIT3 + and S4 CCL21 +). For the first run, we only included the fibroblast and epithelial compartments, whereas in the second run we included the fibroblast, epithelial and immune compartments. The third and final run was conducted using the entire single-cell dataset with global group annotations (e.g. Immune and Muscularis). The single-cell RNA-seq datasets were normalized with Seurat using the NormalizeData function (method = "LogNormalize") and ScaleData. The top 3000 most variable genes were defined as the variable features (FindVariablefeatures, features = 3000). The same normalization procedure was applied to the merged d0 and d14 Visium dataset. AllBoth normalized datasets were then subjected to PCA (RunPCA). Transfer anchors were detected using FindTransferAnchors, setting one of the scRNA-seq datasets as reference and the merged d0 and d14 Visium dataset as query. Cell type prediction scores were estimated using the TransferData function with the previously defined anchor set, setting the cell type labels or group annotations defined by Fawkner-Corbett et al. as the reference data labels and using the PCA reduction object for weight reduction. For each pair of cell type prediction score (at the cell type level) and factor activity vector, a Pearson correlation score was computed to find the correlation structure between cell types and factors, which is shown in Fig. 6b.

**Gene Set enrichment analysis of IBD risk genes in ST factors.** A set of 122 IBD risk genes identified in humans were first converted to their mouse homologs using biomaRt. Out of these 122 genes, 95 mouse homologs were obtained and found to be expressed in the spatially resolved transcriptomics dataset. A correlation matrix was computed using the scaled and normalized expression for the 95 IBD risk genes and the diagonal values were set to 0. From this correlation matrix, a dendrogram was constructed using hclust from the *stats* R package (method = "ward.D2") and 3 clusters were defined by the cutree function (k = 3, *stats* R package). Cluster 3 defined a set of genes with a clear co-expression structure and these genes were further leveraged to compute an IBD risk gene module score (AddModuleScore from *Seurat*). Associations between the IBD risk gene set and factors were determined by conducting Gene Set Enrichment Analysis (GSEA). For each factor (vector of factor activity values across all spots), an enrichment score was calculated for the IBD risk gene set using fgsea from the *fgsea* R package. The only significant association found was for factor 9 (adjusted p-value = 0.012).

**Enrichment analysis of RNAseq derived DSS colitis kinetic modules in spatial transcriptomics data.** To integrate the ST dataset with the kinetic transcriptomic analysis of murine DSS colitis, we used the combined NNMF ST analysis of colon d0 and d14 and the module analysis from[10]. We computed enrichment scores for the gene signatures defining each kinetic module within the gene factor gene loading vectors using fgsea function from the fgsea R package (minSize = 15, maxSize = 500). Shared signatures were defined for module-factor pairs by selecting the intersect of the kinetic module gene set with the top 100 most contributing genes of the factor. To quantify the activity of the shared gene signature in the colonic tissue, the function AddModuleScore from Seurat was used.

**Statistics and reproducibility.** qPCR experiments from Figs. 1g and 3g were performed on three biological replicates. Differences between multiple samples were analyzed by one-way ANOVA followed by Bonferroni's post hoc analysis. Statistical analysis was performed using GraphPad Prism 9.0.1. Representative pictures from the Human Protein Atlas displayed in Figs. 1d and 2c, e are representative of 2–12 pictures. The spatial transcriptomics data presented in all the Main and Supplementary Figures are generated from $n = 1$ mouse on d0 and $n = 1$ mouse on d14 after DSS treatment. A biological replicate of d14 spatial transcriptomic analysis is presented in Supplementary Fig. 8 (with a new NNMF analysis). The micrographs (such as those in Figs. 4b, 5d–g, 6c–e, 7f) are a magnification of a representative slide shown adjacent to them.

**Reporting summary**. Further information on research design is available in the Nature Research Reporting Summary linked to this article.

## Data availability

The datasets generated during this study have been deposited in the Gene Expression Omnibus (GEO) database under accession codes GSE169749 and GSE190595 for spatial transcriptomics, and GSE163638 for scRNAseq. The published datasets used in this study are available under the following accession codes: GSE131032 (mouse longitudinal DSS kinetics);[10] GSE158702 (human scRNAseq)[32]. The pictures from the Human Protein Atlas are available at: https://www.proteinatlas.org/. Source data are provided with this paper.

## Code availability

Code for PROGENy analysis is available at https://github.com/saezlab/visium_colon_si. Code to explore the healing colon ST datasets is available at https://github.com/ludvigla/murine_colon_explorer and https://github.com/ludvigla/healing_intestine_analysis.

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

## Acknowledgements

We thank members of the Villablanca lab for helpful comments and Maria Lim for editorial assistance. We thank Kim Thrane for her invaluable expertise in setting up the Visium experiments and for help with tissue imaging. C.E. was supported by the Marie Sklodowska-Curie grant agreement No 844712. L.L. was funded by grants from the Helmsley foundation. S.D. was supported by Cancerfonden (CAN 2016/1206). E.J.V. was supported by grants from the Swedish Research Council, VR grant K2015-68X-22765-01-6 and 2018-02533, Formas grant nr. FR-2016/0005, Cancerfonden (19 0395 Pj), and the Wallenberg Academy Fellow program (2019.0315). The computations and data handling were enabled by resources provided by the Swedish National Infrastructure for Computing (SNIC) at KTH partially funded by the Swedish Research Council through grant agreement no. 2018-05973. Some schematics were partially created with BioRender.com.

## Author contributions

S.M.P., S.D., A.F., O.E.D., R.M., X.L., G.M. and C.E. performed experiments. L.L., R.O.R., K.P.T. and K.S. performed bioinformatics analysis of the transcriptomic data. E.J.V. and S.D. conceived the idea. N.G., J.S.R. and J.L. provided resources. S.M.P., L.L. and E.J.V. wrote the paper. All authors discussed the data, read, and approved the manuscript.

## Funding

## Competing interests

E.J.V. and N.G. have received research grants from F. Hoffmann-La Roche. C.E., L.L. and J.L. are scientific consultants for 10X Genomics Inc. The remaining authors declare no competing interests.
