## [Peer Review File · Nature Communications]

Reviewers' Comments:

Reviewer #3:

Remarks to the Author:

The manuscript by Parigi et al. demonstrates a very nice application of the 10x Visium array to study local variations in gene expression, signaling pathways and functional structures along the proximal to distal axis of the mouse colon. They also examined the colon's response to chemical injury, as well as mapping known biomarkers associated with human colon disorders.

The use of Harmony to spatially map and compare shared features across the two datasets provided a nice solution to visualizing biological gene expression patterns despite the technical noise between the dataset. I also found the sample mounting (Swiss roll) on the 10x array simple yet innovative. By rolling up the long tubular structure, they maximized the number of usable spots for ST, while preserving all of the spatial information across multiple regions. I also found that many of the functional differences (e.g. lymphoid structures, lamina vs. serous, proximal vs. distal) were just right for the resolution and density of spots in the current generation of 10x Visium, and the authors take advantage of everything ST has to offer, despite its lower spatial resolution. Their data analysis and visualization techniques were adequately described. The manuscript is well-written and edited, and the figures are clearly described and annotated.

Overall, I found this manuscript to be quite interesting both biologically and technically, as well as being of high quality. It demonstrates innovative ways to capture and visualize data, and it provides a potentially useful tool and datasets for mapping the spatial variation of human gut disease markers in model systems. I think the caveats in mapping human disease markers (e.g. UC, IBD) to a mouse model should be discussed in more detail. Regardless, I think the concept of rapidly creating a reference tissue atlas onto which human disease markers can be mapped over time is very nice. I wish I had more critical comments to improve the manuscript, but I am largely happy with the manuscript as written, and I enjoyed reading it.

I do have one major comment. I could not easily find the number of mice, samples, serial sections, or replicates used for this paper. Looking at H&E and the spatial map, it appears that there were two ST datasets. Given their emphasis on biology, I would think that additional biological replicates and possibly time points are needed to make a convincing argument about differentially expressed genes or signaling pathways. I do not know how practical it would be to provide more replicates, but I encourage it. If the dataset were to be used as a reference to map additional features in the future, such replicates would be very reassuring.

Additionally, I have two very minor comments. It would be useful to show scale bars for the spatial stains, maps or diagrams. There are several spelling errors in the supplementary method section.

Reviewer #4:

Remarks to the Author:

In this study, the authors utilized spatial transcriptomics (ST) to define how the transcriptomic landscape is spatially organized in the steady state and healing murine colon. Interestingly, they found spatially organized transcriptional programs defining compartmentalized mucosal healing and regions with dominant wired pathways. Decreased p53 activation defines areas with increased presence of proliferating epithelial stem cells. This work thus provides a publicly available resource defining principles of transcriptomic regionalization of the colon during mucosal healing. The following points need to be addressed.

1. In Fig 1, the authors determined the region-specific mRNA expression of HMGCS2, ANG4, and B4GALT1 at steady condition. Did the authors observe that the expression of these genes was also altered under inflammatory situations? Additionally, (Line 107) NNMF_1 and NNMF_2 were inconsistent with the data in Figure 1C.
2. In Fig 3, the data pointed out that REG3B and MUC2 played a key role in establishing the barrier integrity. As we know, REG3B and MUC2 are classical genes that maintain the intestinal barrier integrity. Did authors find another novel genes according to spatial transcriptomics data? The same point was also seen in Fig 5?

3. In Fig 7, the authors found the critical genes related to the pathways of neutrophil activity. Were there enriched neutrophil activity-related genes defined by functional enrichment analysis using the top contributing genes of factors?
4. Regarding to the spatial transcriptome analysis, when compared to the spatial transcriptome of mouse samples on day 0 and day 14 in the DSS-induced murine colitis model, the mouse samples shall be increased under DSS-induced inflammation on day 7 to clarify the factors that play a key role in the healing process.
5. Non-responsive UC1 patients had more obvious tissue damage and ulcers, while the genes of responsive UC2 patients were evenly distributed on D0 and D14. Could it be considered that the tissue damage and ulcers were mild or even absent? How to explain these findings? In addition, the specific classification and source of the gene sets of UC1 and UC2 patients were not clear.
6. The limitations of spatial transcriptome analysis need to be addressed, compared with single-cell sequencing and proteomics.

Point-by-point response

We sincerely thank the Reviewers for their valuable comments and suggestions. We are pleased that the reviewers have acknowledged the novelty and significance of our study. We have revised our manuscript accordingly and provided a detailed point-by-point response to the comments below. All relevant passages in the revised text have been highlighted in yellow.

REVIEWER COMMENTS

Reviewer #3 (Remarks to the Author):

The manuscript by Parigi et al. demonstrates a very nice application of the 10x Visium array to study local variations in gene expression, signaling pathways and functional structures along the proximal to distal axis of the mouse colon. They also examined the colon's response to chemical injury, as well as mapping known biomarkers associated with human colon disorders.

The use of Harmony to spatially map and compare shared features across the two datasets provided a nice solution to visualizing biological gene expression patterns despite the technical noise between the dataset. I also found the sample mounting (Swiss roll) on the 10x array simple yet innovative. By rolling up the long tubular structure, they maximized the number of usable spots for ST, while preserving all of the spatial information across multiple regions. I also found that many of the functional differences (e.g. lymphoid structures, lamina vs. serous, proximal vs. distal) were just right for the resolution and density of spots in the current generation of 10x Visium, and the authors take advantage of everything ST has to offer, despite its lower spatial resolution. Their data analysis and visualization techniques were adequately described. The manuscript is well-written and edited, and the figures are clearly described and annotated.

Overall, I found this manuscript to be quite interesting both biologically and technically, as well as being of high quality. It demonstrates innovative ways to capture and visualize data, and it provides a potentially useful tool and datasets for mapping the spatial variation of human gut disease markers in model systems. I think the caveats in mapping human disease markers (e.g. UC, IBD) to a mouse model should be discussed in more detail. Regardless, I think the concept of rapidly creating a reference tissue atlas onto which human disease markers can be mapped over time is very nice. I wish I had more critical comments to improve the manuscript, but I am largely happy with the manuscript as written, and I enjoyed reading it.

Response: We sincerely thank the Reviewer for these laudatory remarks. We are particularly pleased that our work was appreciated from a technical and conceptual standpoint. We are also gratified that the manuscript was well presented and thus expectantly accessible by a broad audience.

I do have one major comment. I could not easily find the number of mice, samples, serial sections, or replicates used for this paper. Looking at H&E and the spatial map, it appears that there were two ST datasets. Given their emphasis on biology, I would think that additional biological replicates and possibly time points are needed to make a convincing argument about differentially expressed genes or signaling pathways. I do not know how practical it would be to provide more replicates, but I encourage it. If the dataset were to be used as a reference to map additional features in the future, such replicates would be very reassuring.

Response: We thank the reviewer for bringing up this point. Our main observations in the manuscript are: 1) the existence of a molecular regionalization at steady state condition that is partly altered during mucosal healing; and 2) during mucosal healing, there are distinct biological processes (e.g. response to damage, keratinization) that co-occur in distinct areas of the regenerating tissue.

In our original submission, we have provided qPCR data on biological triplicates to validate two of the main observations in our manuscript, which are the regionalization at steady state condition (Fig 1f) and the distal expression of genes normally expressed in the proximal colon occurring during mucosal healing (Fig 3f).

However, to strengthen our observation of distinct biological processes co-occurring during healing and as per the reviewer request, we have now included an additional ST dataset of the colonic tissue at d14. NMF analysis using d0 and the two replicates at d14 (mucosal healing) show that processes like “keratinization” (factor_11’, new Supplementary Fig. 8), “danger response” (factor_15’, new Supplementary Fig. 8), “regeneration” (factor_17’, new Supplementary Fig. 8) emerged at d14 compared to d0 in two distinct d14 datasets. However, as expected, it is important to note that the localization of these processes varies between samples (in line with the variability and patchy appearance of DSS-induced colitis). Nonetheless, these processes nicely matched with the underlying histology of the respective tissues and are molecularly comparable among the replicates.

We have included the following conclusions in the text at page 9, line 258-265: “Analysis of a second d14 dataset showed similar results as processes associated with “keratinization” (new Factor_11’), “danger response” (new Factor_15’), and “regeneration” (new Factor_17’) consistently emerged during mucosal healing (Supplementary Fig. 8). Overall, DSS-induced injury resulted in an assorted co-occurrence of different histopathological processes within the murine colon. Furthermore, our analysis revealed a previously unappreciated heterogeneous transcriptional and regional landscape of tissue repair (Fig. 4e), which varies in location among biological replicates (Supplementary Fig. 8).”

Additionally, I have two very minor comments. It would be useful to show scale bars for the spatial stains, maps or diagrams.

Response: Thanks to the reviewer for pointing this out. We have now included scale bars to all figures.

There are several spelling errors in the supplementary method section.

Response: We have now revised and corrected spelling errors in the supplementary method section.

Reviewer #4 (Remarks to the Author):

In this study, the authors utilized spatial transcriptomics (ST) to define how the transcriptomic landscape is spatially organized in the steady state and healing murine colon. Interestingly, they found spatially organized transcriptional programs defining compartmentalized mucosal healing and regions with dominant wired pathways. Decreased p53 activation defines areas with increased presence of proliferating epithelial stem cells. This work thus provides a publicly available resource defining principles of transcriptomic regionalization of the colon during mucosal healing.

Response: We sincerely thank the Reviewer for his/her valuable comments, which we believe will improve the quality and accessibility of our study.

The following points need to be addressed.

1. In Fig 1, the authors determined the region-specific mRNA expression of HMGCS2, ANG4, and B4GALT1 at steady condition. Did the authors observe that the expression of these genes was also altered under inflammatory situations?

Response: We analyzed these genes during mucosal healing and we observed that while *Hmgsc2* macroscopically retained the same relative distribution as on d0, the intensity and expression pattern for *Ang4* and *B4Galt1* seems to be different/altered compared to d0 (see Figure 1 for Reviewer 4). Although we consider this an interesting observation, we do not have a reasonable explanation to motivate these findings and to explain why these two genes are altered, while for instance *Hmgsc2* retains the same expression pattern as in steady state. Thus, we feel that the interpretation may go beyond the scope of the analysis in Figure 1 (regionalization at steady state conditions).

Figure 1 for Reviewer 4. Spatial distribution of the normalized expression of the genes Hmgsc2, Ang4 and B4galt1 in steady state (d0, on the left) and in two replicates of healing colon (d14, on the right). Each dot represents a ST spot and is color-coded based on an enrichment score (high: red, low: grey).

Additionally, (Line 107) NNMF_1 and NNMF_2 were inconsistent with the data in Figure 1C.

Response: We thank the reviewer for noticing this mistake. We have now corrected it in the revised version.

2. In Fig 3, the data pointed out that REG3B and MUC2 played a key role in establishing the barrier integrity. As we know, REG3B and MUC2 are classical genes that maintain the intestinal barrier integrity. Did authors find another novel genes according to spatial transcriptomics data? The same point was also seen in Fig 5?

Response: We thank the reviewer for raising this point. To address the concern about Figure 3, we have now included a supplementary table with all the differentially expressed genes in each cluster (supplementary Table 1). With this list, we aim to provide a resource to identify and validate novel candidates that might be involved in mucosal healing based on their cluster's spatial distribution.

Regarding the second question ("The same point was also seen in Fig5?"), we are afraid that we couldn't completely understand the Reviewer's question. In Figure 5, we show the distribution of the 14 pathways deposited in the pathway method PROGENy and their correlation with the Factors from NNMF analysis. Since those 14 pathways are the only ones that were curated and included in the PROGENy pipeline so far, it is not

possible to include new pathways in the analysis from Fig 5. To help overcome this limitation, we provided GO pathway analysis of the Factors in Fig.4. If we misinterpreted the Reviewer's question, we kindly ask the Reviewer to help us addressing his/her issues in the next round of revision.

3. In Fig 7, the authors found the critical genes related to the pathways of neutrophil activity. Were there enriched neutrophil activity-related genes defined by functional enrichment analysis using the top contributing genes of factors?

4. Regarding to the spatial transcriptome analysis, when compared to the spatial transcriptome of mouse samples on day 0 and day 14 in the DSS-induced murine colitis model, the mouse samples shall be increased under DSS-induced inflammation on day 7 to clarify the factors that play a key role in the healing process.

5. Non-responsive UC1 patients had more obvious tissue damage and ulcers, while the genes of responsive UC2 patients were evenly distributed on D0 and D14. Could it be considered that the tissue damage and ulcers were mild or even absent? How to explain these findings? In addition, the specific classification and source of the gene sets of UC1 and UC2 patients were not clear.

Response: We thank the reviewer for these pertinent questions. The kinetic DSS-induced colitis dataset was obtained from our previous work (Czarnewski, Nat Comm, 2019) in which we performed a longitudinal transcriptomic analysis of the process of DSS-induced intestinal inflammation and recovery. The longitudinal transcriptomic analysis enabled us to identify modules which were associated with distinct biological processes (based on GO terms using top expressing genes), such as inflammation (m1) and mucosal healing (m6). In addition, comparing the mouse dataset to the transcriptomic profiles of ulcerative colitis (UC) patients enabled us to identify conserved genes between species and stratify UC patients into two categories (i.e. UC1 and UC2).

Here, we investigated the association between the ST Factors and the modules defined by the longitudinal analysis in mouse (from Czarnewski, Nat Comm, 2019) (Fig 7b) and we mapped the genes that characterize m1 and m6 onto our ST datasets (Fig 7c and d). Finally, we mapped the gene signature characterizing UC1 and UC2 patients (from Czarnewski, Nat Comm, 2019) in our ST data set (Fig 7e).

To clarify the origin and characteristics of the UC1/UC2 transcriptomic signatures previously defined (from Czarnewski, Nat Comm, 2019), we included a new panel showing a schematic representation of these datasets in new figure 7. We thank the reviewer for helping us to improve figure 7 and hope that the reviewer's question has been clarified.

Answer to point (3): The Reviewer highlights that one of the top signatures of the inflammatory module 1 is neutrophil degranulation (as seen in Fig 7c ii). Considering the overlap between module 1 and Factor 9, we ran gene ontology (GO) functional enrichment analysis on the top genes defining Factor 9. While neutrophil activity-related terms were not enriched in our analysis, we found that factor 9 is associated more broadly with immune system regulation/activation/proliferation (please find below Figure

2 for Reviewer 4). This finding is in line with the top GO enrichment term of module 1 (i.e. inflammatory response). As we believe that this additional analysis will not significantly alter the conclusions drawn in Figure 7, we decided to include it for Reviewer’s evaluation only. However, if the Reviewer feels that it should be included in the revised version of the manuscript, we will include a Supplementary Figure in the next round of revision.

Figure 2 for Reviewer 4. Functional enrichment analysis (Gene Ontology Biological Processes, GO:BP) based on the top genes defining factor 9.

Answer to point (4): The Reviewer raises an important point. By analyzing a d7 sample we could pinpoint Factors that are specific for healing and not inflammation. Our integration of the mouse longitudinal analysis (from Czarnecki et al.) and the ST data set tries to partially address the Reviewer’s question. In Figure 7, we show that m1 genes are increased at d7 and then downregulated by d14. We ran those genes in d14 ST datasets with the idea to show where the remnant “m1 inflammatory genes” were expressed in the tissue. Our conclusion is that m1 inflammatory genes are highly expressed in ILF and damage/repair areas. Based on the Reviewer’s suggestion, we predict that these genes would be broadly expressed (or increased) in ST d7. Unfortunately, we do not have a d7 ST sample and we believe that including such a data set would be beyond the scope of this manuscript and excessively broaden its message. While we agree that it would be an interesting analysis, the goal of our study is to provide a resource characterizing the heterogeneity of tissue healing when compared to steady state. However, we acknowledge the Reviewer’s comment and we consider it as part of the discussion (page 16, line 505-507) and as reported below: “Whether the spatial transcriptomic signature during the acute phase (i.e. DSS d7) would be associated with an expansion of the damage-associated factors identified at d14 remains to be addressed. Nonetheless, the asynchronous nature of the healing

process observed at d14 may be associated with varying degrees of exposure to the external environment and the elaborate architecture of the tissue.”

Answer to point (5): We apologize for the lack of clarity in defining the specific classification and source of the gene sets of UC1 and UC2 patients. Towards this, we now included the citation to the original reference in Figure 7e. In addition, we expanded the relative text at page 13, line 393-396 (and reported below) to better explain the source of UC1 and UC2 signatures.

“Toward this end, we used the recent gene signature identifying ulcerative colitis (UC) subgroups of patients: UC1 and UC2¹⁰. In this previous study¹⁰, by integrating human IBD and murine DSS transcriptomic datasets, we identified UC1 and UC2 as two subgroups of IBD patients characterized by comparable histological disease severity but different underlying genetic signature. This molecular classification is clinically relevant...”

In the original manuscript from Czarnewski et al., UC1 and UC2 patients were identified by overlapping human transcriptomic datasets of IBD patients onto the longitudinal mouse transcriptomic dataset generated during DSS-colitis. Of note, UC1 and UC2 patients were macroscopically comparable from an histological standpoint, as seen by their similar Mayo Score. However, their genetic signature and response to therapy was different. Here, in Figure 7e, we mapped the gene signatures defining UC1 and UC2 patients onto the mouse d14 ST data set generated in this study and found enrichment of UC1 signature in the damage area. This result suggested that UC1 gene profile is more associated with tissue damage compared to UC2 gene signature. As the reviewer pointed out, one possibility is that UC1 patients have more subclinical tissue damage and ulcers (that is not detectable by routine endoscopy) compared to UC2. To acknowledge this possibility the following statement was added in the discussion (at page 17, line 544-546):

“Our data suggest that UC1 patients have increased tissue damage compared with UC2 patients, which might contribute to their poor response to biological therapies.”

6. The limitations of spatial transcriptome analysis need to be addressed, compared with single-cell sequencing and proteomics.

We have now included a new paragraph (lines 525-535) in the discussion section, in which we discussed limitations of the technology compared to scRNAseq and proteomics. The new text reads:

Limitations

Among limitations when using spatial transcriptomics technology is that each observation (spot) typically represents the averaged expression profile from multiple cells. For the 10x Visium technology, the diameter of a spot is 55 microns, meaning that mixtures of 1-10 cells are profiled simultaneously. Here, we demonstrate how these

mixed profiles can be deconvolved by NMF to identify transcriptional programs in a broader tissue context. The characteristics of these transcriptional programs are further supported by morphological detail at single-cell resolution obtained from HE bright field images. However, we anticipate that integration with complementary technologies with higher resolution, such as scRNA-seq, would deepen our understanding of the cellular networks and communication pathways underlying these transcriptional programs. Although ST is currently limited to the analysis of mRNA, we anticipate that multimodal spatial profiling will help pinpoint pivotal events in tissue regeneration. For example, parallel targeting of proteins would not only provide a more comprehensive molecular map of the colon, but be particularly useful to identify cell-cell crosstalk events identified by ligand-receptor profiling. This information could help us elucidate crosstalk between epithelial and immune cells which have been implicated in the orchestration of the wound healing process.

Reviewers' Comments:

Reviewer #3:

Remarks to the Author:

I thank the authors for taking the time to address my questions. The addition of an extra dataset to replicate the intra-tissue heterogeneity that was seen on Visium, as well as the explanation of confirmatory qPCR data from multiple replicates were helpful.

Reviewer #4:

Remarks to the Author:

This manuscript has been revised thoughtly, and all questions were also addressed accordingly.